# ⟨X⟩-Coder: Advancing Competitive Programming with Fully Synthetic Tasks, Solutions, and Tests

## Abstract

Competitive programming presents great challenges for Code LLMs due to its intensive reasoning demands and high logical complexity. However, current Code LLMs still rely heavily on real-world data, which limits their scalability. In this paper, we explore a fully synthetic approach: training Code LLMs with entirely generated tasks, solutions, and test cases, to empower code reasoning models without relying on real-world data. To support this, we leverage feature-based synthesis to propose a novel data synthesis pipeline called *SynthSmith*. SynthSmith shows strong potential in producing diverse and challenging tasks, along with verified solutions and tests, supporting both supervised fine-tuning and reinforcement learning. Based on the proposed synthetic SFT and RL datasets, we introduce the *X-Coder* model series, which achieves a notable pass rate of 62.9 avg@8 on LiveCodeBench v5 and 55.8 on v6, outperforming DeepCoder-14B-Preview and AReal-boba²-14B despite having only 7B parameters. In-depth analysis reveals that scaling laws hold on our synthetic dataset, and we explore which dimensions are more effective to scale. We further provide insights into code-centric reinforcement learning and highlight the key factors that shape performance through detailed ablations and analysis. Our findings demonstrate that scaling high-quality synthetic data and adopting staged training can greatly advance code reasoning, while mitigating reliance on real-world coding data. Our code, data and models will be made publicly available.

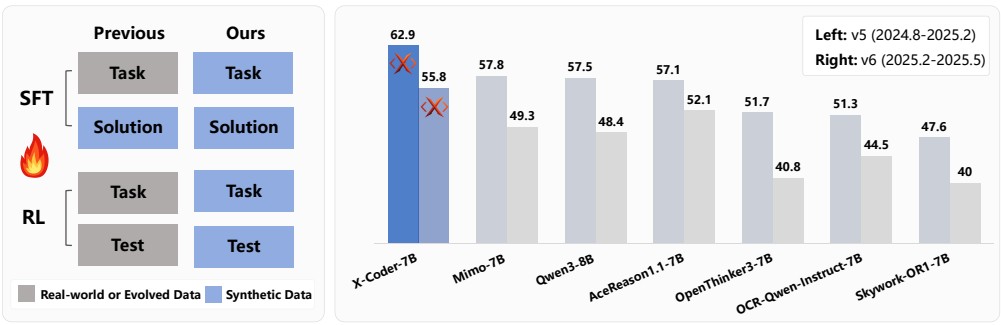

Figure 1: Left: SynthSmith generates high-quality synthetic tasks, solutions, and test cases to support both SFT and RL training. Right: Avg@8 results on LiveCodeBench. X-Coder achieves significant performance gains on competitive programming using *fully* synthetic data.

## 1 Introduction

As code language models advance, reasoning-focused models such as OpenAI-o1-ioi (OpenAI et al., 2025) have reached expert-level performance in programming. Classic benchmarks including HumanEval (Chen et al., 2021; Liu et al., 2023a) and MBPP (Austin et al., 2021) have been largely

solved, whereas tasks from LiveCodeBench (Jain et al., 2024) and Codeforces continue to demand deeper reasoning and more complex algorithmic problem solving.

Recently, DeepSeek-R1 (Guo et al., 2025) has opened two opportunities for further boosting the reasoning capabilities of Code LLMs. The first is supervised fine-tuning (SFT) (Ouyang et al., 2022) on long Chain-of-Thought (CoT) demonstrations to distill reasoning patterns into student models (Hugging Face, 2025; Labs, 2025; Liu et al., 2025a). The second is reinforcement learning (RL) (Schulman et al., 2017) with GRPO (Shao et al., 2024b) and related algorithms to refine reasoning foundation models (Luo et al., 2025; Fu et al., 2025; He et al., 2025).

Both pathways have proven effective but face a common bottleneck: progress on competitive programming remains constrained by the scarcity of datasets. Widely used collections such as APPS (Hendrycks et al., 2021), CodeContests (Li et al., 2022), and TACO (Li et al., 2023) are heavily reused during post-training. They remain too modest in scale to support continued benefits and still lack the level of sufficiently challenging, diverse, and scalable. Meanwhile, collecting new real-world data tailored for competitive programming is also challenging. Although recent work has synthesized rewritten or evolutionary variants (Luo et al., 2024; Liu et al., 2025a; Xu et al., 2025) from existing resources, their diversity and complexity remain tightly bounded by the seed tasks.

To address this gap, we explore a fully synthetic approach: training Code LLMs with fully generated tasks, solutions, and test cases. Building on this insight, we present *SynthSmith*, a novel coding data synthesis pipeline tailored for competitive programming. To enable the synthesis of diverse and challenging competitive programming tasks, SynthSmith extends feature-based methods (Wang et al., 2025) with competition-oriented feature extraction, dedicated feature integration, and multi-style task construction. SynthSmith further supports the development of high-quality solutions and tool-based test case generation, both of which are cross-validated through the proposed dual-verification strategy. Thereby, SynthSmith demonstrates strong potential in producing scalable and challenging tasks, together with verified solutions and tests, offering support for both SFT and subsequent RL. Starting from a base model (e.g., Qwen3-8B-Base) or a non-reasoning model (e.g., Qwen2.5-Coder-7B-Instruct), we present the X-Coder series, which achieves significant performance gains on challenging LiveCodeBench v5 and v6 without relying on any real-world data, as shown in Figure 1. Beyond this, built upon verl (Sheng et al., 2025), we present an RL infrastructure featuring automated high-concurrency code validation, leveraging the CPUs of all distributed machines to support efficient and large-scale code execution.

Our in-depth analysis examines (i) whether synthetic SFT data scale effectively and which dimensions scale more favorably; (ii) the role of code-centric reinforcement learning, including the "good-gets-better" principle and RL's resilience to noisy supervision; (iii) the factors that shape performance (long- vs. short-CoT, effects of solution verification, task style, and data-selection strategies); and (iv) the bottlenecks that limit code reasoning, together with the chained relationship among task difficulty, reasoning length, and pass rate. We further conduct case studies to uncover cognitive behaviors that emerge after SFT and RL, including reward hacking and undesirable patterns.

We make the following contributions:

(1) We explore a fully synthetic approach and propose a novel data synthesis pipeline tailored for competitive programming, producing high-quality datasets for both SFT and RL stages.

(2) We train both base and non-reasoning LLMs under an SFT-then-RL paradigm to develop the X-Coder model series, which achieves significant performance gains on LiveCodeBench v5 (avg@8: 62.9) and v6 (avg@8: 55.8), along with extensive analyses and ablations.

(3) We introduce an optimized infrastructure for code RL, featuring a dedicated sandbox environment that speeds up code execution and improves training efficiency.

## 2 SYNTHSMITH: SYNTHESIS OF COMPETITION-LEVEL CODING DATA

We introduce SynthSmith, a fully synthetic framework for constructing competitive programming tasks that support both the SFT and RL stages. Figure 2 illustrates the SynthSmith pipeline, which consists of four key steps: (i) generating novel and challenging problems (with the capacity for easy scaling in quantity); (ii) constructing diverse and comprehensive input test cases for each problem (including boundary and stress tests); (iii) producing high-quality candidate solutions; and (iv) em-

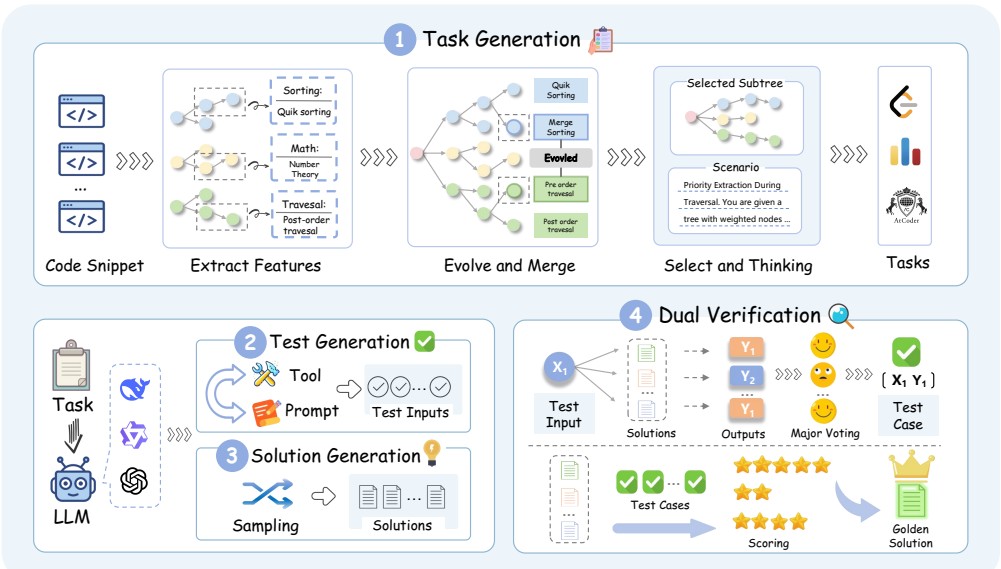

Figure 2: **Framework of SynthSmith.** SynthSmith first extracts and evolves competitive programming related features from small-scale code instruction data and merges them into tree structures. It then samples subtrees from the feature tree, selects a compatible feature set, and formulates a scenario that naturally integrates these consistent features. Novel tasks are generated based on a proposed scenario according to specific styles. Advanced reasoning models are used to synthesize solutions and tests for the generated tasks, which are further cross-verified using the proposed dual-verification strategy to yield reliable test outputs and the top solution.

ploying a dual-verification strategy that cross-checks solutions with test cases to yield more accurate test outputs and more reliable solutions.

**(i) Task Generation.** Inspired by EpiCoder (Wang et al., 2025), which generates novel programming tasks through a feature-based framework by combining sampled features into problem scenarios, we extend this approach with three key improvements to synthesize diverse and complex tasks tailored for competitive programming. First, instead of relying on broad definitions of features, we explicitly extract and evolve competition-related features from 10k code snippets in the TACO dataset (Li et al., 2023) using GPT-4o-0513 (detailed in §B.1). Second, formulating competitive scenarios from a rich feature tree is non-trivial, as LLMs often oversimplify complex prompts into trivial cases, thereby reducing both diversity and difficulty. To address this, we adopt a two-stage process that separates feature selection from scenario formulation: first, selecting mutually consistent features for meaningful composition; and second, formulating hint-free tasks that demand genuine reasoning. We further incorporate one-shot prompting to improve task understanding and instruction-following. Third, we adapt the synthesis method to support multi-style task generation, covering Codeforces[1]-style tasks (standard input/output with imaginable narrative contexts), LeetCode-style[2] tasks (starter code with predefined function signatures), and AtCoder[3]-style tasks (concise specifications with minimal explanations), thereby enhancing task diversity. Examples of the task generation process are provided in §B.2, together with difficulty estimates on generated tasks in §B.3.

**(ii) Test Input Generation.** Obtaining sufficient and accurate test cases is a formidable challenge. Problems from competitive programming platforms often do not provide test cases, or only provide a limited number, due to platform constraints. This results in insufficient quantity, difficulty, and coverage of test cases during RL training. To address the inherent scarcity of test cases in synthesized data, we investigate two complementary methods for generating the input component of the test case. The *Prompting-based* method instructs the LLM to interpret the problem's input constraints

---

[1] https://codeforces.com/
[2] https://leetcode.com/
[3] https://atcoder.jp/

and directly generate multiple test inputs, covering both standard cases and edge-case instances. The *Tool-based* method leverages CYaRon[4], a dedicated test case generation library, enabling the LLM to construct test inputs by invoking functions documented within the library after understanding the problem. For each task, we generate a set of $n$ test case inputs $[x_1, x_2, \ldots, x_n]$. Detailed description of test input generation is provided in §D, and a comparative analysis is presented in Sec 4.

**(iii) Candidate Solutions Generation.** For each task, we generate multiple candidate solutions using advanced open-source reasoning LLMs, obtaining $m$ answers $[A^1, A^2, \ldots, A^m]$. We verify that each candidate solution includes a complete reasoning process and a Python code implementation, and we ensure the absence of syntax errors through static analysis methods based on Abstract Syntax Tree (AST). Filtering criteria are provided in §C.1.

**(iv) Dual-Verification of Solutions and Test Cases.**

To ensure the robustness and reliability of both the generated solutions and the constructed test cases, we adopt a dual-verification strategy. Step 1 of this strategy extends the principle of self-consistency (Wang et al., 2023) by applying majority voting across candidate solutions from multiple LLMs, which mitigates model-specific biases and enhances generalization, thereby yielding a reliable test output for each input. Step 2 then identifies the top-performing candidate solution by incorporating test case difficulty weighting alongside a hold-out validation set.

*Step 1: Verification of Test Cases via Consensus Voting.* First, we establish a preliminary ground truth for each test case input. For a given input $x_i$, we execute all candidate solutions to obtain a set of outputs $\{y_i^1, y_i^2, \ldots, y_i^m\}$, where $y_i^j = A^j(x_i)$. A provisional ground truth output $\hat{y}_i$ is determined via majority voting:

$$\hat{y}_i = \underset{y}{\arg\max} \sum_{j=1}^{m} \mathbb{I}(y_i^j = y) \quad , \tag{1}$$

where $\mathbb{I}(\cdot)$ is the indicator function. This yields a candidate test set $\mathcal{T}_{candidate} = \{(x_1, \hat{y}_1), \ldots, (x_n, \hat{y}_n)\}$. Crucially, we posit that not all test cases are of equal importance; boundary or edge cases are critical for robust evaluation. We therefore introduce a weighting function $w(x_i) \to w_i$ that assigns a higher score to more challenging test cases. The weight $w_i$ is determined by a set of heuristics based on input characteristics, such as character or token count, structural complexity, or semantic novelty, which serve as proxies for difficulty.

*Step 2: Verification of Solutions via Weighted Evaluation and Hold-out Validation.* To ensure that our selected "golden" solution generalizes beyond the generated data, we partition the candidate test set. We randomly sample a subset of $\mathcal{T}_{candidate}$ (e.g., 50%) to form a hold-out validation set, $\mathcal{T}_{val}$. The remaining data constitutes our primary weighted test suite, $\mathcal{T}_{golden}$. The dual-verification process culminates in selecting the golden answer, $A_{golden}$. A candidate solution $A^j$ is first evaluated on $\mathcal{T}_{golden}$ using a weighted score. The top-performing candidate, $A'_{golden}$, is identified as:

$$A'_{golden} = \underset{A^j \in \{A^1, \ldots, A^m\}}{\arg\max} \sum_{(x_i, \hat{y}_i) \in \mathcal{T}_{golden}} w_i \cdot \mathbb{I}(A^j(x_i) = \hat{y}_i) \quad . \tag{2}$$

The final confirmation of $A_{golden}$ is contingent upon its performance on the unseen hold-out set $\mathcal{T}_{val}$. We verify that $A'_{golden}$ also achieves the highest (or a competitively high) unweighted accuracy on $\mathcal{T}_{val}$ relative to other candidates. This additional validation step ensures that the selected solution is not merely overfitted to the specifics of the weighted test cases but demonstrates superior, generalizable correctness. The detailed algorithm is provided in §E.

Finally, we obtain $A_{golden}$ and $T_{golden}$ for each task $q$. The pair $[q, A_{golden}]$ is used to compute the SFT loss, and $[q, T_{golden}]$ are used for RL via the GRPO algorithm.

**Discussion.** Compared to rewriting-based data synthesis methods (Luo et al., 2024; Liu et al., 2025a), SynthSmith reduces reliance on seed tasks by formulating novel tasks from evolved competitive features. Compared with EpiCoder, it generates more challenging tasks and selects high-quality solutions via a dual-verification strategy, yielding a 21% absolute performance gain on Live-CodeBench v5 (Figure 5c). Moreover, SynthSmith extends data synthesis to the RL stage, showing that synthetic RL data can further improve performance beyond the SFT model as shown in Table 1.

---

[4]https://github.com/luogu-dev/cyaron

Table 1: Performance on LiveCodeBench v5. X-Coder shows strong coding expertise with fewer, fully synthetic tasks, and achieves additional gains through subsequent RL stages. †: OpenThinker3 integrates human-written tasks with synthetic math tasks. rStar-Coder augments real-world coding tasks with synthesized rewrites for mixed training, whereas X-Coder relies on fully synthetic tasks.

| Model | Base Model | SFT | RL | Size | Data | Task | Metric | V5 Score | V6 Score |
|-------|------------|-----|-----|------|------|------|--------|----------|----------|
| **SFT Baselines** | | | | | | | | | |
| Bespoke-Stratos (Labs, 2025) | Qwen2.5-Instruct (Qwen et al., 2025) | ✓ | ✗ | 7B | 17k | Real | pass@1 | 16.2 | 8.57 |
| OpenThinker3 (Guha et al., 2025) | Qwen2.5-Instruct | ✓ | ✗ | 7B | 1,200k | Mixed$^\dagger$ | - | 51.7 | 40.8 |
| OlympicCoder (Hugging Face, 2025) | Qwen2.5-Coder-Instruct (Hui et al., 2024) | ✓ | ✗ | 7B | 100k | Real | - | 40.9 | 19.3 |
| OCR-Qwen-Instruct (Ahmad et al., 2025) | Qwen2.5-Instruct | ✓ | ✗ | 7B | 736k | Real | avg@64 | 51.3 | 44.5 |
| rStar-Coder (Liu et al., 2025a) | Qwen2.5-Coder-Instruct | ✓ | ✗ | 7B | 580K | Mixed$^\dagger$ | avg@16 | 57.3 | – |
| Qwen3-8B (Yang et al., 2025) | Qwen3-8B-Base | ✓ | ✗ | 8B | - | Real | - | 57.5 | 48.4 |
| **RL Baselines** | | | | | | | | | |
| Skywork-OR1 (He et al., 2025) | R1-Distilled-Qwen (DeepSeek-AI, 2025) | ✗ | ✓ | 7B | 124k | Real | avg@32 | 47.6 | 40.0 |
| DeepCoder-Preview (Luo et al., 2025) | R1-Distilled-Qwen | ✗ | ✓ | 14B | 24k | Real | pass@1 | 57.9 | 48.5 |
| AReal-boba² (Fu et al., 2025) | R1-Distilled-Qwen | ✗ | ✓ | 14B | 24k | Real | avg@32 | 58.1 | 56.7 |
| **SFT-then-RL Baselines (Stage 1)** | | | | | | | | | |
| AceReason1.1-SFT (Liu et al., 2025b) | Qwen2.5-Math (Yang et al., 2024) | ✓ | ✗ | 7B | 2.2M | Real | avg@8 | 51.2 | - |
| MiMo-SFT (Xiaomi et al., 2025) | MiMo-Base | ✓ | ✗ | 7B | 500k | Unclear | avg@8 | 52.3 | 45.5 |
| Klear-Reasoner-SFT (Su et al., 2025) | Qwen3-Base (Yang et al., 2025) | ✓ | ✗ | 8B | 1500k | Real | avg@8 | 58.5 | 49.6 |
| X-Coder-Qwen2.5-SFT | Qwen2.5-Coder-Instruct | ✓ | ✗ | 7B | 200k | Syn | avg@8 | **60.3**$_{\pm2.5}$ | **53.5**$_{\pm1.7}$ |
| X-Coder-Qwen3-SFT | Qwen3-8B-Base | ✓ | ✗ | 8B | 200k | Syn | avg@8 | **59.4**$_{\pm2.0}$ | **55.4**$_{\pm2.3}$ |
| **SFT-then-RL Baselines (Stage 2)** | | | | | | | | | |
| AceReason1.1 | AceReaon1.1-SFT | ✓ | ✓ | 7B | - | Real | avg@8 | 57.2 | 52.1 |
| MiMo | MiMo-SFT | ✓ | ✓ | 7B | 130k | Unclear | avg@8 | 57.8 | 49.3 |
| Klear-Reasoner | Klear-Reasoner-SFT | ✓ | ✓ | 8B | 106k | Real | avg@8 | 61.6 | 53.1 |
| X-Coder-Qwen2.5 | X-Coder-Qwen2.5-SFT | ✓ | ✓ | 7B | 40k | Syn | avg@8 | **62.9**$_{\pm1.8}$ | **55.8**$_{\pm1.9}$ |
| X-Coder-Qwen3 | X-Coder-Qwen3-SFT | ✓ | ✓ | 8B | 40k | Syn | avg@8 | **64.0**$_{\pm2.5}$ | **56.5**$_{\pm1.3}$ |

## 3 EXPERIMENT

**Setup.** In this study, we adopt GPT-o3-mini (OpenAI, 2025) for task formulation, Deepseek-R1-0528 (DeepSeek-AI, 2025) and Qwen3-235B-A22B-Thinking-2507 (Yang et al., 2025) for solution sampling, and R1-0528 for test case generation. Statistics for SFT datasets are provided in §C.2. For SFT, we set the learning rate at 5e-5, with a global batch size of 128 to train 8 epochs. For RL, the reward is defined as the fraction of passed tests among all given tests (detailed in §A.2). The program executes in an isolated sandbox environment deployed with Redis, which supports optimized concurrent code testing (infrastructure details are provided in §A.5). Training configurations and costs are supplemented in §A.4.

**Evaluation.** We evaluate Code LLMs on LiveCodeBench (Jain et al., 2024) v5 (covering problems released between Aug. 2024 and Feb. 2025) and v6 (Feb. to May 2025), which are the most widely used benchmarks for code reasoning models. Baselines are documented in §A.6. To ensure a fair comparison, we use Qwen2.5-Coder-7B-Instruct and Qwen3-8B-Base as backbones, and report the avg@8 pass rate using a sampling temperature of 0.6 with top-p 0.95 to align with the baselines.

### 3.1 MAIN RESULTS

As shown in Table 1, during the SFT stage, X-Coder-SFT achieves an avg@8 pass rate of 60.3. Compared with RL baselines, X-Coder-SFT exhibits a clear advantage over 14B-based RL models (e.g., DeepCoder-Preview-14B, AReal-boba²-14B), despite those models being built on the stronger foundation R1-Distilled-Qwen. Relative to SFT-then-RL models, X-Coder further boosts its performance after RL, reaching 62.9. On Qwen3-Base, X-Coder attains an avg@8 pass rate of 64.0.

### 3.2 SFT EXPERIMENTS AND ANALYSIS

During the SFT stage, we investigate a central question: Can the SFT dataset be effectively scaled, and along which dimension should it be scaled more favorably? To explore this, we are inspired by AceReason-Nemotron 1.1 (Liu et al., 2025b) and expand the SFT dataset from two distinct perspectives: increasing the number of unique tasks and enlarging the number of solutions per task. We design seven subsets (v1–v6): v1–v4 increase the number of unique tasks (32k, 64k, 128k, and 192k unique prompts, each with 1 solution), while v5–v6 expand the number of solutions per task (16k unique prompts with 4 solutions, and 8k unique prompts with 8 solutions). The results in Figure 3 reveal a promising scaling trend, where v4 > v3 > v2 > v1, with performance steadily improving from 43.7% to 62.7%.

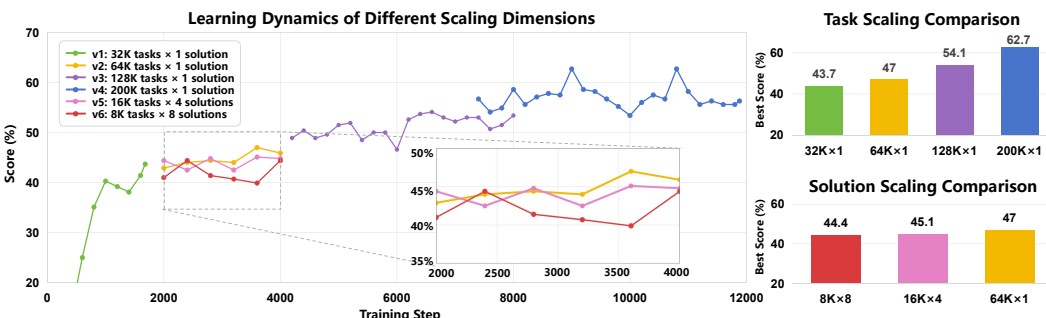

Figure 3: Scaling laws on the SFT dataset generated by SynthSmith. Left: Performance comparison of on LiveCodebench v5 to examine scaling trend. Right: Performance comparison across scaling unique tasks and scaling solutions per task.

Furthermore, the comparison v2 (64k×1) > v5 (16k×4) > v6 (8k×8) shows that scaling the number of unique tasks is more effective than increasing the number of solutions per task. When computational budget is fixed, expanding task diversity is more efficient for improving generalization.

**Comparison with Real-World and Synthetic Code Datasets.** For real-world datasets, we compare against OpenCodeReasoning (Ahmad et al., 2025), the largest reasoning-based synthetic dataset to date for competitive coding. We train our dataset and OpenCodeReasoning using the same number of training tokens with Qwen2.5-Coder-7B-Instruct. The results are shown in Table 2. Our proposed dataset yields a 6.7-point improvement after SFT, with most gains coming from the medium and hard splits. The improvement is attributed to our pipeline's ability to synthesize more challenging tasks, which demand longer reasoning (average length 17.7k vs. 8.0k), and to provide greater prompt diversity, which proves more effective than increasing solution diversity.

Table 2: OpenCodeReasoning vs. Dataset from SynthSmith.

| Model | Avg. | Easy | Medium | Hard |
|---|---|---|---|---|
| OCR-Qwen-7B-Instruct (Ahmad et al., 2025) | 51.3 | 95.4 | 64.0 | 18.0 |
| OCR-Qwen-Coder-7B-Instruct | 53.6 | 95.2 | 67.0 | 21.8 |
| X-Coder-Qwen-Coder-7B-Instruct | **60.3** (+6.7) | **96.8** | **73.3** | **37.8** |

Table 3: Synthetic Data by SelfCodeAlign vs. by SynthSmith.

| Method | Task Gen. | Ans. Gen. | Data | Score |
|---|---|---|---|---|
| SelfCodeAlign (Wei et al., 2024) | GPT-o3-mini | DeepSeek-R1 | 10k | 27.1 |
| SynthSmith (Ours) | GPT-o3-mini | DeepSeek-R1 | 10k | **31.7** (+4.6) |

For synthetic datasets, we implemented the SelfCodeAlign (Wei et al., 2024) method using same teacher models and adapted it to the competitive programming domain to deliver a 10k-sample dataset. The results in Table 3 shows our method achieves a 4.6 performance gains, demonstrating the effectiveness of our data synthesis strategy for competitive programming.

### 3.3 RL EXPERIMENTS AND ANALYSIS

Our investigation of the RL stage uncovers the following key insights into its role and behavior:

**(i) RL as a Powerful Refiner**. RL fine-tuning is not merely an incremental add-on but a powerful optimization step. As shown in Table 1, when applied to a converged SFT model using only code data, it yields a substantial 4.6% absolute gain in average pass-rate. This highlights RL's unique capability to refine policy beyond the distribution of the initial supervised dataset.

**(ii) The "Good-gets-Better" Principle**. RL performance is tightly coupled to the strength of the SFT initializer. Using two SFT models trained on similar data distributions but with different Live-

CodeBench scores as starting points, we observe in Figure 4 that, under identical RL settings, the stronger initializer consistently attains higher rewards.

A stronger SFT foundation enablings to explore a more promising policy space and achieve a higher performance ceiling. This underscores the importance of a high-quality initial model as a prerequisite for effective RL.

**(iii) Resilience to Noisy Supervision**. Contrary to the common assumption that RL requires pristine reward signals, our experiments reveal a resilience to data imperfections during RL. The model also effectively benefits from synthetic test cases, suggesting that RL can be successfully deployed in scenarios with large-scale but imperfect feedback (Wang et al., 2020; Lv et al., 2025), significantly lowering the barrier to code RL data collection.

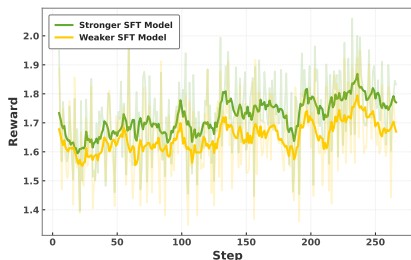

Figure 4: Reward comparison of weak and strong SFT models as RL initializer.

## 4 ABLATION STUDY

Despite the strong performance of X-Coder, the determinants of high-quality synthetic data for SFT remain insufficiently understood. To elucidate these factors, we conduct a comprehensive ablation along six axes: (i) the effect of the proposed dual-verification strategy; (ii) the impact of distinct thinking types in solutions; (iii) the influence of task styles; (iv) a head-to-head comparison of tasks produced by SynthSmith versus those from open-source synthetic datasets; (v) data-selection strategies to identify patterns that shape downstream performance; and (vi) comparison of prompting-based and tool-based test generation strategies.

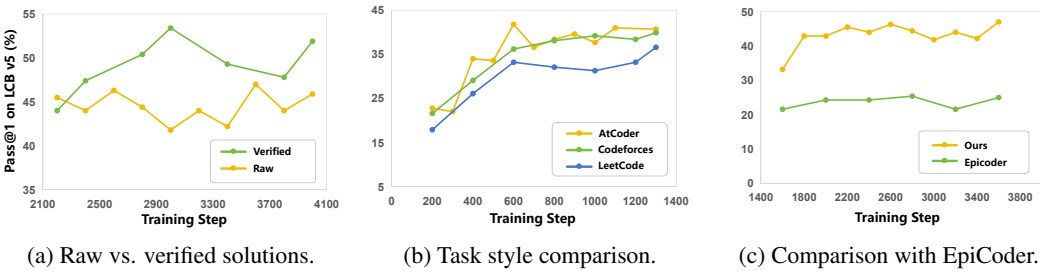

(a) Raw vs. verified solutions.    (b) Task style comparison.    (c) Comparison with EpiCoder.

Figure 5: Ablations on verification, task style, and task sources.

**Q1: Dual-verification for High-Quality Data Curation.** To mitigate the noise introduced by stochastically sampled solutions, we employ a dual-verification strategy for data curation. This strategy first leverages the self-consistency principle to identify the most likely correct solution from multiple candidates. Subsequently, these candidate solutions are executed against a comprehensive set of test cases to verify their functional correctness and robustness, thereby capturing subtle runtime errors (e.g., ValueError, IndexError, or Timeout) that are undetected by static analysis methods like AST checks. The efficacy of this approach is validated by our empirical results, as shown in Figure 5a. Using an identical backbone (Qwen2.5-Coder-7B-Instruct) and dataset (64k tasks), the model trained on verified solutions significantly outperforms its counterpart trained on raw solutions. However, this quality assurance comes at a considerable computational cost. For instance, fully verifying 200k samples necessitates the generation of 1.6 million long-CoT trajectories and 24 million test executions. This overhead establishes a clear trade-off, as prior work (Li et al., 2025; Gandhi et al., 2025) indicates that models can still learn effectively from unverified long-CoT data, making raw-solution training a more resource-efficient, albeit potentially less performant, alternative.

**Q2: Solution Types: Long CoT vs. Short CoT.** The length of CoT proves to be a critical factor for performance, with longer CoTs yielding superior results despite higher training costs. To demonstrate this, we compare the Qwen2.5-Coder-7B-Instruct trained on solutions generated by DeepSeek-R1-0528 (Long-CoT) and Qwen3-235B-A22B-Instruct-2507 (Short-CoT) for an identical set of tasks (200k).

As shown in Table 4, the long-CoT approach achieves a 17.2% absolute gain. This substantial improvement justifies the increased computational demand, which manifests as a slower convergence requiring 8–10 epochs compared to the 2–3 epochs needed for short-CoT data.

Table 4: Long CoT vs. Short CoT.

|  | Epoch | LCB v5 | LCB v6 |
|---|---|---|---|
| Short-CoT | 3 | 35.0 | 29.3 |
|  | 8 | 43.1 | 37.6 |
|  | Δ | +8.1 | +8.3 |
| Long-CoT | 3 | 42.9 | 36.0 |
|  | 8 | **60.3** | **53.5** |
|  | Δ | +17.4 | +17.5 |

**Q3: Ablation on Task Style.** We evaluate the effect of task styles (AtCoder, Codeforces, and LeetCode) by synthesizing three corpora of 32k tasks each (8k unique problems with 4 solutions per problem) from identical input features. For each corpus, solutions are generated with DeepSeek-R1-0528 and used to fine-tune the Qwen2.5-Coder-7B-Instruct. Results are shown in Figure 5b. Although AtCoder-style tasks yield slightly higher scores, we adopt Codeforces-style as the predominant format in our demonstration dataset (Codeforces : AtCoder : LeetCode = 70 : 15 : 15), reflecting its prominence as the mainstream competitive-programming platform.

**Q4: Tasks from SynthSmith vs Tasks from EpiCoder-380k.** We randomly select 64k tasks from our SFT dataset and another 64k from EpiCoder-380k, and use DeepSeek-R1-0528 to complete solutions. Figure 5c shows that tasks from SynthSmith yield a 21% absolute performance gain, demonstrating its ability to produce high-quality tasks tailored for competitive programming.

**Q5: Data Selection.** To investigate data utilization efficiency, we explore task selection strategies for competitive programming. Specifically, we evaluate three approaches: (1) difficulty-based selection, where GPT-4o-2411 assigns discrete difficulty scores to tasks, simulating the Codeforces rating system; (2) rationale-based selection, where DeepSeek-R1-0528 generates CoT reasoning for each task, and tasks that elicit longer reasoning traces are prioritized; and (3) random selection as a baseline. For validation, each strategy independently samples a 50k-task subset from a 200k-task pool. Solutions are generated by Qwen3-235B-A22B-Instruct-2507, and models were trained for three epochs with a 16k context length.

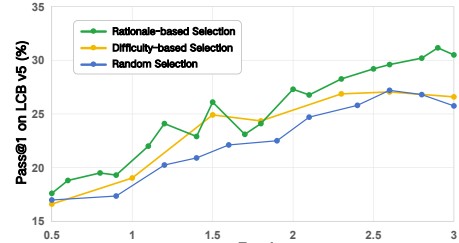

Figure 6: Comparison of data selection.

As shown in Figure 6, tasks that induce longer CoT are regarded as more valuable training data for competitive programming, as they demand deeper reasoning and are potentially more challenging.

**Q6: Prompting-based vs. Tool-based Test Generation.** We compare prompting-based and tool-based test generation using tasks from CodeContests (Li et al., 2022). We leverage the corresponding golden solutions to evaluate the accuracy and complexity of the tests produced by the two approaches. The results in Table 5 show that the tool-based approach outperforms the prompting-based method across multiple dimensions. Qualitatively, it is more versatile, capable of systematically generating random, scalable, boundary, and stress tests, which are essential for robust code evaluation but not supported by prompting-based methods.

Quantitatively, the tool-based approach achieves a higher pass rate on ground-truth solutions (87.9% vs. 77.4%), confirming that its test cases are more accurate and reliable. It also generates more challenging and discriminative tests, as reflected by the lower consensus ratio (78.8% vs. 82.0%), which indicates stronger effectiveness in uncovering subtle bugs. In addition, the tool-based generator provides broader test coverage, albeit at a higher computational cost.

Table 5: Comparison of Prompting-based and Tool-based Test Generation. The tool-based approach excels in test diversity, accuracy, and the ability to generate more challenging test cases.

|  | Random | Scalable | Boundary | Stress | Cost | Avg Tests | Min Tests | Max Tests | Consensus | Pass Rate |
|---|---|---|---|---|---|---|---|---|---|---|
| Prompting-based | ✗ | ✗ | ✗ | ✗ | low | 13.6 | 5 | 15 | 82.0% | 77.4% |
| Tool-based | ✓ | ✓ | ✓ | ✓ | high | 18.3 | 5 | 27 | 78.8% | 87.9% |

## 5 DISCUSSION

In this section, we present an in-depth analysis of the main challenges in code reasoning. Reasoning models often suffer from assertion errors, highlighting persistent reasoning limitations on harder tasks. We further identify a mediation pattern among task difficulty, reasoning length, and pass rate, and extend our analysis with test-time scaling experiments and case studies on cognitive behavior, reward hacking, and undesirable patterns.

**Failure Analysis.** We classify failure cases into seven types: Wrong Answer (output mismatches the expected answer), Time Limit Exceeded, Memory Limit Exceeded, No Code Block Generated (truncated due to heavy reasoning before the final code is generated), Incomplete Code Block (partial code without closure), Function Signature Mismatch (incorrect function definition), and Syntax Error (complete code with syntax issues). The error distribution in Table 6 indicates that the primary bottleneck lies in reasoning capability, with most errors stemming from wrong answers. Two other major failure categories are No Code Block Generated and Time Limit Exceeded (TLE). We carefully inspected the no-code samples and found that all of them exceeded the 32k context window, causing the reasoning process to be truncated and incomplete. The frequency of TLE errors highlights the need for Code LLMs to prioritize code efficiency.

After RL, X-Coder reduces assertion errors compared to its SFT counterparts by learning from correctness-based rewards. At the same time, the RL optimization process may introduce instability, leading to issues such as syntax errors, signature mismatches, and other flaws.

Table 6: Distribution of failure cases for 16 rollouts on LiveCodeBench v5 (268 tasks).

| Error Type | Qwen2.5-Coder-7B-Instruct | Qwen3-8B | X-Coder-7B-SFT | X-Coder-7B |
|---|---|---|---|---|
| Wrong Answer | $194.6 \pm 10.7$ | $87.1 \pm 4.6$ | $69.6 \pm 3.7$ | $67.9 \pm 4.9$ |
| No Code Block | $6.5 \pm 8.2$ | $7.7 \pm 1.2$ | $21.9 \pm 3.7$ | $11.8 \pm 3.9$ |
| Time Limit Exceeded | $18.1 \pm 4.1$ | $21.8 \pm 3.8$ | $13.7 \pm 3.3$ | $11.5 \pm 2.6$ |
| Memory Limit Exceeded | $0.0 \pm 0.0$ | $0.0 \pm 0.0$ | $0.0 \pm 0.0$ | $0.17 \pm 0.4$ |
| Incomplete Code Block | $0.0 \pm 0.0$ | $0.0 \pm 0.0$ | $0.0 \pm 0.0$ | $1.0 \pm 0.8$ |
| Signature Mismatch | $0.0 \pm 0.0$ | $0.0 \pm 0.0$ | $0.0 \pm 0.0$ | $1.0 \pm 0.8$ |
| Syntax Error | $0.0 \pm 0.0$ | $0.0 \pm 0.0$ | $0.0 \pm 0.0$ | $8.3 \pm 2.2$ |

**Pass Rate by Reasoning Token Length.** The results in Table 7 show that the pass rate decreases sharply as reasoning token length increases, exhibiting a clear downward trend. This finding runs counter to the intuitive expectation that greater test-time token usage reflects deeper reasoning and should therefore yield higher accuracy. Instead, we observe a significant chained relationship among problem difficulty, reasoning length, and pass rate: problem difficulty is positively correlated with reasoning length, while reasoning length is strongly negatively correlated with pass rate. This mediation pattern can be summarized as higher difficulty → longer reasoning length → lower pass rate.

Table 7: Performance analysis by reasoning token length.

| Token | Total | Passed | Easy | Medium | Hard |
|---|---|---|---|---|---|
| 0–5k | 38 | 38 | 30/30 (100.0%) | 8/8 (100.0%) | 0/0 (–) |
| 5k–10k | 41 | 38 | 16/17 (94.1%) | 14/16 (87.5%) | 8/8 (100.0%) |
| 10k–15k | 41 | 32 | 10/11 (90.9%) | 14/19 (73.7%) | 8/11 (72.7%) |
| 15k–20k | 52 | 36 | 4/4 (100.0%) | 16/16 (100.0%) | 16/32 (50.0%) |
| 20k–25k | 36 | 15 | 1/1 (100.0%) | 9/13 (69.2%) | 5/22 (22.7%) |
| >25k | 60 | 10 | 0/0 (–) | 2/14 (14.3%) | 8/46 (17.4%) |
| **Total** | **268** | **169** | **61/63 (96.8%)** | **63/86 (73.3%)** | **45/119 (37.8%)** |

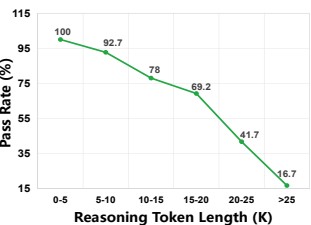

Figure 7: Pass rate by token.

**Test-time Scaling.** We compare the pass@k performance of Qwen2.5-Coder-7B-Instruct, Qwen3-8B, X-Coder-7B-SFT, and X-Coder-7B in Figure 8. X-Coder-7B outperforms its foundation model by 51.3 points in pass@16, and matches Qwen3-8B with 8× fewer rollouts. Moreover, X-Coder shows a larger gap between pass@1 and pass@16 compared to Qwen3-8B (19.2 vs. 13.8), indicating greater diversity in the reasoning patterns it can explore. Although RL models begin with higher initial performance than the SFT model, the gap does not expand within 16 rollouts, suggesting that RL improves pass@1 but may not escape its starting point (Wu et al., 2025).

**Behaviors after SFT and RL.** After SFT, the model frequently exhibits cognitive behaviors such as planning, verification, backtracking, and reflection, as illustrated by the case study in §H.1. This suggests that such behaviors can be directly distilled from the teacher rather than induced by the RL process. During the later stages of RL, the model shows signs of reward hacking, attempting to exploit edge cases for partial rewards instead of producing genuine solutions, as detailed in §H.3. We also observe several bad patterns in code reasoning, including premature termination when the model is aware that the context is running out, recalling memorized submissions in C++

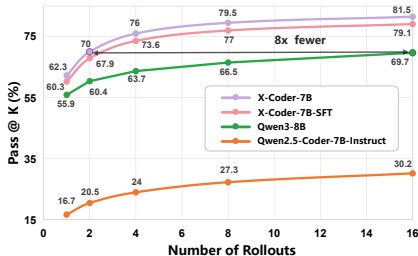

Figure 8: Test-time performance.

and attempting to translate them into Python, and emitting incomplete code before the context window is exhausted. These cases are illustrated in §H.2.

## 6    RELATED WORK

**Data Synthesis for Code.** The research community has long recognized the scarcity of high-quality coding tasks. To address this, Wizard-Coder (Luo et al., 2024) extends Evol-Instruct (Xu et al., 2024) by evolving basic code-instruction data into augmented variants. rStar-Coder (Liu et al., 2025a) further adapts this augmentation strategy to the competitive programming domain. CodeEvo (Sun et al., 2025) introduces a coder–reviewer interaction framework to collaboratively synthesize high-quality instruction–code pairs.

SelfCodeAlign (Wei et al., 2024) advanced task synthesis beyond simple seed evolution by introducing concept composition. It extracts fundamental concepts from seed problems and uses these building blocks to generate a vastly larger bank of novel coding tasks through the systematic combination of underlying concepts, pioneering a new pathway for scaling problem synthesis. Epicoder (Wang et al., 2025) follows this direction by sampling sub-features from a large and expressive feature tree to formulate novel problems, further improving task complexity and diversity.

This work targets competitive programming, a domain where previous synthetic methods struggle to synthesize tasks requiring deep reasoning and accurate test cases. We demonstrate that a fully synthetic pipeline offers a practical and scalable solution to these challenges. Methodologically, we employ competition-oriented feature extraction to synthesize challenging, coherent, and diverse tasks. Crucially, we construct high-fidelity test cases using prompt- and tool-based input generation combined with voting-based labeling. These accurate tests enable *Code RL* advancements.

**Post-training Recipe for Code Reasoning Model.** From the training perspective, current approaches to building coding-expert LLMs generally fall into three paradigms: (i) purely supervised fine-tuning on real-world tasks or their rewritten or evolved variants (Labs, 2025; Guha et al., 2025; Liu et al., 2025a), (ii) purely reinforcement-based fine-tuning using a GRPO-related (Shao et al., 2024b; He et al., 2025; Luo et al., 2025; Fu et al., 2025) algorithm, and (iii) reinforcement learning staged after supervised fine-tuning on mixed coding and mathematical data (Liu et al., 2025b; Xiaomi et al., 2025; Su et al., 2025). High-quality code data is scarcer than mathematical data. Consequently, existing approaches rely heavily on real-world data and lack a stable two-stage recipe for coding expertise, often mixing in mathematics with little evidence of success on code alone. In this paper, we show that stable and consistent improvements in code reasoning can be achieved solely with synthetic data, while also reducing the risk of data leakage shown in §G.

## 7    CONCLUSION

In this paper, we explore a fully synthetic approach to competitive programming and propose a novel data synthesis framework that demonstrates how synthetic tasks, solutions, and tests can train large reasoning models to achieve significant performance gains, thereby reducing reliance on real-world data. Building on this framework, we contribute scalable synthetic SFT and RL training sets, supported by a dedicated RL infrastructure, and introduce the X-Coder series. Furthermore, we provide insights into code-centric SFT-then-RL training, ablate key factors that shape performance, and present in-depth analyses with illustrative case studies of code reasoning models.

## ETHICS STATEMENT

This work aims to advance large code reasoning models for competitive programming through fully synthetic data. No personal, private, or sensitive information is included in the datasets or experiments, and no ethical risks are associated with this study.

## REPRODUCIBILITY STATEMENT

With respect to reproducibility, we affirm our commitment to ensuring that all reported results can be faithfully reproduced, and we will provide the necessary resources and documentation to facilitate replication. The anonymous repository link for reference and reproduction is https://anonymous.4open.science/r/x-coder.

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

# Appendix

## A  TRAINING AND EVALUATION

### A.1  SFT-THEN-RL TRAINING

**Supervised Fine-tuning.** Given a dataset of task–solution pairs $\mathcal{D} = \{(x_i, y_i)\}_{i=1}^N$, the model with parameters $\theta$ is trained by minimizing the negative log-likelihood (NLL) of the target solution $y$ conditioned on the task $x$:

$$J_{\text{SFT}}(\theta) = -\mathbb{E}_{(x,y)\sim\mathcal{D}}\left[\sum_{t=1}^{|y|} \log \pi_\theta\big(y_t \mid x, y_{<t}\big)\right]. \tag{3}$$

The loss is applied over full long-CoT trajectories, including both reasoning steps and final code, enabling the model to imitate not only the solutions but also the underlying reasoning patterns.

**Reinforcement Learning.** Proximal Policy Optimization (PPO) (Schulman et al., 2017) is a widely adopted policy gradient method in Reinforcement Learning from Human Feedback (RLHF) (Christiano et al., 2017) for LLM due to its balance between exploration and exploitation and its empirical robustness. The method optimizes a policy $\pi_\theta$ by using a clipped surrogate objective to limit policy divergence, incorporating a value function to estimate expected rewards, and an entropy term to encourage exploration. The overall objective function for PPO is designed to maximize the policy performance while maintaining stability, and it is typically formulated as minimizing the following:

$$J_{\text{PPO}}(\theta) = \mathbb{E}_{s\sim P(S), a\sim\pi_\theta(a|s)}\left[\min\left(\frac{\pi_\theta(a|s)}{\pi_{\theta_{\text{old}}}(a|s)}A(s,a),\ \text{clip}\left(\frac{\pi_\theta(a|s)}{\pi_{\theta_{\text{old}}}(a|s)}, 1-\epsilon, 1+\epsilon\right)A(s,a)\right)\right] \tag{4}$$

where the expectation is computed over states $s$ (drawn from distribution $P(S)$) and actions $a$ (sampled from the current policy $\pi_\theta(a \mid s)$), combining the minimum of two terms: (1) the product of the probability ratio $\frac{\pi_\theta(a|s)}{\pi_{\theta_{\text{old}}}(a|s)}$ and the advantage function $A(s,a)$, where the advantage function quantifies the relative benefit of taking action $a$ in state $s$; and (2) the same product but with the probability ratio clipped to the interval $[1-\epsilon, 1+\epsilon]$. Here, $\epsilon$ is a hyperparameter governing the magnitude of policy updates. This clipping mechanism effectively constrains excessive policy changes, thereby enhancing training stability.

However, its application to LLMs encounters significant challenges, including substantial computational overhead from maintaining a critic network, which increases memory usage and training time for models with billions of parameters. Additionally, training stability can be undermined by inaccurate value function estimates or suboptimal tuning of Generalized Advantage Estimation (GAE) (Schulman et al., 2016) parameters, issues that become more pronounced as LLMs scale in size. To address these limitations, Group Relative Policy Optimization (GRPO) (Shao et al., 2024a) has emerged as an efficient alternative. By eliminating the critic network, GRPO reduces computational and memory demands, estimating advantages directly from rewards of multiple rollouts to the same prompt, thus leveraging the comparative nature of reward models and offering a scalable solution for LLM training. The GRPO objective function is mathematically formulated as an averaged composite expression across multiple rollouts, incorporating policy ratio optimization and KL regularization:

$$J_{\text{GRPO}}(\theta) = \frac{1}{G}\sum_{i=1}^{G}\frac{1}{|a_i|}\sum_{t=1}^{|a_i|}\left\{\min\left(\rho_{i,t}\hat{A}_{i,t},\ \text{clip}(\rho_{i,t}, 1-\epsilon, 1+\epsilon)\hat{A}_{i,t}\right) - \beta D_{\text{KL}}[\pi_\theta \parallel \pi_{\text{ref}}]\right\} \tag{5}$$

where $\rho_{i,t} = \frac{\pi_\theta(a_{i,t}|s, a_{i,<t})}{\pi_{\theta_{\text{old}}}(a_{i,t}|s, a_{i,<t})}$ denotes the probability ratio of the old and new strategies. $G$ is the number of rollouts per prompt, $|a_i|$ denotes the length of the $i$-th action sequence, $\hat{A}_{i,t}$ estimates the advantage of action $a_{i,t}$ at timestep $t$. The clipping is analogous to PPO, and $\beta$ penalizes deviations from $\pi_{\text{ref}}$ via the KL-divergence term. The objective averages across rollouts and timesteps, combining a clipped probability ratio (to stabilize updates while leveraging advantage signals) with a KL penalty to balance policy improvement against alignment with the reference policy. This dual mechanism ensures controlled optimization by restricting drastic policy shifts while maintaining coherence with prior behavior.

## A.2 REWARD FUNCTION.

We remove formatting rewards (e.g., enforcing "think" tags), as the SFT model already follows the format, allowing the policy to focus on passing test cases. Given a rollout, the reward $R$ is practiced as:

$$\mathcal{R} = \begin{cases} -2, & \text{if no code is extracted or the code fails to compile,} \\ 0, & \text{if the code compiles but passes no test cases,} \\ \dfrac{5.0 \times \#\text{passed}}{\#\text{total}}, & \text{otherwise.} \end{cases} \tag{6}$$

We adopt a continuous reward setting, as it provides denser supervision than the all-or-nothing alternative and leads to faster convergence (Wei et al., 2025; Dai et al., 2024).

## A.3 TRAINING DYNAMICS.

As shown in Figure 9 and Figure 10, we present the SFT training curves (loss and token accuracy). Figure 11 and Figure 12 illustrate the RL training curves (reward and entropy).

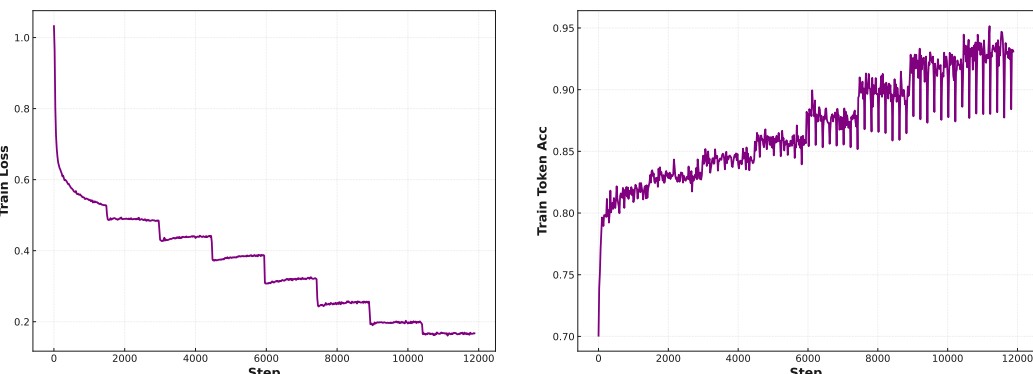

Figure 9: Training loss of SFT.
Figure 10: Training token accuracy of SFT.

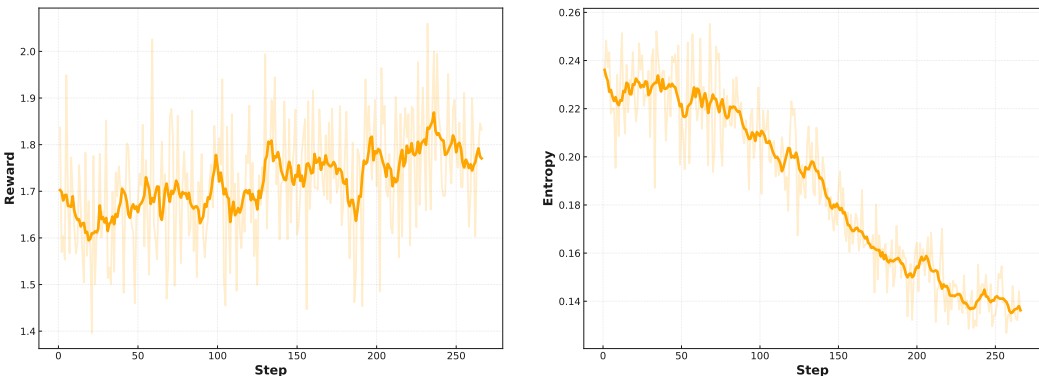

Figure 11: Training reward of RL.
Figure 12: Training entropy of RL.

## A.4 TRAINING CONFIGS AND COSTS

For SFT, we use a learning rate of 5e-5 with a global batch size of 128 for 8 training epochs. For RL, the policy models are updated with a global batch size of 128 and a consistent learning rate of 7e-5, without applying the KL-divergence constraint to the starter model, and employ a rollout temperature of 1.0 with 8 rollouts to encourage exploration.

Training large reasoning models incurs significant costs compared to standard (eg. short-CoT) instruction models. In the SFT stage, the dominant overhead stems from longer sequence lengths and the need for more update epochs, which together lead to several times more compute consumption than training non-reasoning counterparts. In the RL stage, the major bottleneck lies in generating multiple rollouts for each problem used for GRPO-algorithm.

Concretely, training X-Coder on Qwen2.5-Coder-7B-Instruct required 128 H20 Enterprise (96 GB) GPUs for 220 hours during SFT, and 32 H200 (141 GB) GPUs for 7 days to complete 270 update steps during RL. We are going to make X-Coder a readily accessible, open-source model, enabling the community to benefit from its capabilities without having to bear the training costs.

### A.5 A DISTRIBUTED FRAMEWORK FOR AUTOMATED CODE VERIFICATION

To provide a robust and scalable solution for code validation, we develop a distributed arbitration framework inspired by open-source repository implementations[5]. The system is based on a microservice architecture, comprising a *FastAPI*-based asynchronous API Gateway, a pool of code execution workers in the sandbox and a central *Redis* instance. *Redis* serves as a high-performance message broker and state manager, effectively decoupling the client-facing gateway from the back-end computational workers. This architectural choice facilitates independent scaling, deployment, and enhances the overall resilience of the system. **Based on this evaluation framework, we implemented highly concurrent code testing during RL training.** We used batching when submitting tasks to the *Redis* server to achieve high concurrency even with low request rates. This process required the server to distribute all test tasks to different workers, utilizing the CPU power of all participating machines. Figure 13 shows the system diagram of the framework.

The framework's efficacy is derived from its strategic implementation of *Redis* data structures. Task distribution is managed by a *Sorted Set*, which functions as a time-prioritized FIFO queue; submissions are added with a timestamp score via *ZADD*, and workers atomically retrieve the next task using *BZPOPMIN*. This approach ensures ordered processing and prevents race conditions. For result transmission, each task is assigned a dedicated *List*, to which a worker pushes the outcome using *RPUSH*. The API Gateway then performs a blocking pop (*BLPOP*) on this unique list to retrieve the corresponding result efficiently. Furthermore, worker health and presence are monitored using *String* keys with a Time-To-Live (TTL). Workers periodically refresh their key's TTL as a heartbeat, enabling the system to automatically detect and de-register unresponsive nodes.

The resulting system exhibits several key advantages. The asynchronous, in-memory nature of its core components yields high throughput and low-latency performance. Its design is inherently scalable, as the stateless worker pool can be expanded horizontally to meet computational demand, while native support for *Redis* Cluster addresses data-tier bottlenecks. Finally, the framework's reliability is bolstered by the atomicity of *Redis* operations and the integrated fault-detection mechanism, ensuring dependable and consistent code verification.

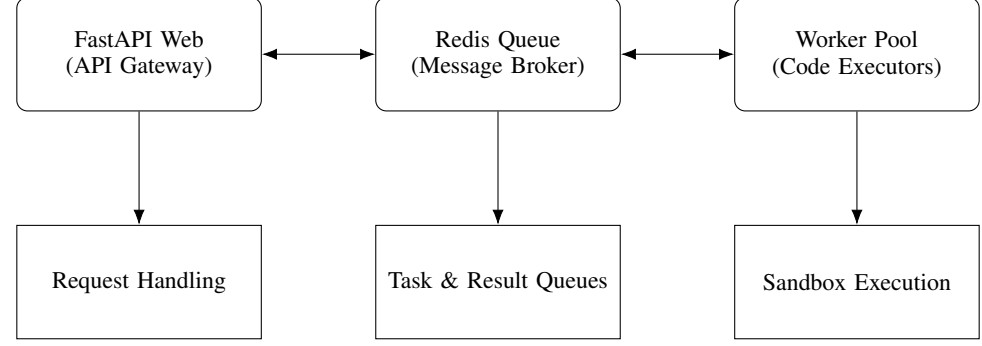

Figure 13: The distributed architecture of the code verification framework.

---

[5]https://github.com/0xWJ/code-judge.git

## A.6 BASELINES

We compare the X-Coder with three categories of baselines: (1) SFT model, e.g., Bespoke-Stratos, OlympicCoder, OCR-Qwen-Instruct, OpenThinker3, Qwen3-8B, and rStar-Coder; (2) RL model, including Skywork-OR1, DeepCoder-14B-Preview, and AReal-boba²-14B; (3) SFT-then-RL model, such as AceReason1.1, Klear-Reasoner, and MiMo-RL.

## B NOVEL TASK SYNTHESIS

Building on EpiCoder, which synthesizes programming tasks through feature-based combinations, we introduce three key improvements to generate more diverse and complex instructions.

*First*, rather than relying on broad feature definitions, we explicitly extract and evolve competition-related features from 10,000 question–solution pairs in TACO (Li et al., 2023) using GPT-4o-0513 (§B.1). *Second*, we adopt a two-stage process: selecting mutually consistent features and then formulating challenging, hint-free tasks (§B.2). *Third*, we extend the synthesis method to support multi-style generation, covering CodeForces-style tasks (rich narratives with standard I/O), LeetCode-style tasks (starter code with fixed signatures), and AtCoder-style tasks (concise specifications), thereby enhancing task diversity. In §B.3, we further estimate the difficulty of synthesized problems using a trained discriminator.

### B.1 FEATURE EXTRACTION AND EVOLUTION

While EpiCoder extracts general-purpose features from raw corpus, we explictly extract and evol compatetitive programming-related feature. Spefcilily, we design multiple aspect of features that highly relates to competitve programming, such as data structure, algorithm, mathmatical, ect.

We improve the extraction process to guide the LLM to focus on competitive programming–related concepts, as follows:

```
Extract features from the provided problem and solution code related to algorithmic
programming, competitive programming, Leetcode, and Codeforces, following the requirements
for each category below, formatted in JSON structure.

Responses in the following categories should be concise and organized in a JSON format
surrounded with <begin> and <end>. Categories may include nested structures if applicable.
Here is an example of the expected format:

<begin>{
    "programming language": [
        "Python"
    ],
    "problem type": [
        "graph traversal"
    ],
    "algorithm": {
        "graph algorithms":[
            "Dijkstra's algorithm",
            "DFS",
            "BFS"
        ],
        "dynamic programming":[
            "Longest Increasing Subsequence",
            "Knapsack Problem"
        ]
    },
    "data structures": [
        "array",
        "linked list",
        "heap",
        "segment tree"
    ],
    "implementation logic":["recursive", "iterative"]
}<end>

Categories to extract:
1. Programming Language: Note the specific programming language used. Example: ["Python",
"C++"].
2. Problem Type: Outline the type of problem the code is solving. Example: ["graph
traversal", "sorting", "dynamic programming"].
```

```
3. Algorithm: Identify the specific algorithm or method being used in the code. This category
can include the following subcategories:
    3.1 Graph Algorithms: Specify graph algorithms used. Example: ["Dijkstra's algorithm",
"DFS", "BFS"].
    3.2 Sorting Algorithms: Specify sorting algorithms used. Example: ["QuickSort",
"MergeSort"].
    3.3 Dynamic Programming: Specify dynamic programming techniques. Example: ["Longest
Increasing Subsequence", "Knapsack Problem"].
    3.4 Search Algorithms: Identify search algorithms used. Example: ["Binary Search",
"Linear Search"].
    3.5 Other relevant subcategories...
4. Data Structures: Describe the primary data structures utilized. Example: ["array",
"graph", "tree", "heap"].
5. Implementation Logic: Describe the implementation logic. Example: ["iterative",
"recursive", "bit manipulation"].
6. Complexity Analysis: Provide time and space complexity of the code if available. Example:
["Time Complexity: O(n log n)", "Space Complexity: O(n)"]
7. Optimization Techniques: Specify any optimizations applied. Example: ["memoization",
"greedy approaches", "bitwise operations"].
8. Purpose: What the code is used to do. Example: "To find the shortest path in a graph using
Dijkstra's algorithm."
9. Summary: Provide a concise summary. Example: "Solves the given competitive programming
problem using a depth-first search approach to traverse the graph."

Extract as many features as possible and try not to let a feature appear in multiple
categories at the same time.
```

Then we increase the diversity and complexity through evolution along both the breadth and depth dimensions. For example, along the breadth dimension, given an extracted feature such as quicksort, the LLM may evolve new features like bubble sort, even if they were not originally extracted. Along the depth dimension, a concept such as prefix sum can evolve into more advanced variants like difference array or Fenwick tree, reflecting increasing levels of abstraction and difficulty. The overall evolution process is illustrated below.

```
Feature Tree Evolution Task:
You are provided with a feature tree represented as a nested JSON structure. Each node in
this tree represents a feature or a sub-feature of competitive algorithm programming, with
the leaves being the most specific features. Your task is to expand this feature tree both in
depth and breadth. Depth expansion means adding more specific sub-features to existing
leaves. Breadth expnasion means adding more sibling features at the current levels.

Here are some explanations of the features:
{explanations}

The input feature tree will be provided in JSON format, and your output should be a JSON
structure that represents the expanded feature tree.

Output Format:
- Expanded Feature Tree: Provide the expanded feature tree as a JSON structure. Surround the
json with <begin> and <end>.

Input Feature Tree Example:
{
    "algorithm": {
        "sorting": ["quick sort", "merge sort"],
        "tree traversal": ["in-order traversal"]
    },
    "mathematics": [
        "number theory",
        "combinatorics"
    ]
}

Expanded Feature Tree Example:
<begin>
{
    "algorithm": {
        "sorting": {
            "quick sort": ["3-way quick sort", "dual-pivot quick sort"],
            "merge sort": ["top-down merge sort", "bottom-up merge sort"],
            "heap sort":[]
        },
        "tree traversal": {
            "in-order traversal": ["recursive in-order traversal", "iterative in-order
traversal"],
            "pre-order traversal":[],
            "post-order traversal":[],
            "level-order traversal":[],
```

```
             }
         },
     "mathematics": {
         "number theory": [
             "prime factorization",
             "greatest common divisor",
             "power modular reduction"
         ],
         "combinatorics": [
             "Pascals triangle",
             "permutations and combinations",
             "binomial coefficients"
         ]
     }
}
<end>

Constraints:
1. For breadth expansion, add at least 2 new sibling features to each existing node.
2. For deep expansion, you need to add new sub-features to it, provided that you think the
current leaf node has a more fine-grained feature.
3. Focus on generating new and innovative features that are not present in the provided
examples.
4. The features are related to competitive algorithm programming.
Please follow the above constraints and expand the feature tree accordingly.

Input:
{features}

Output:
<begin>expanded feature tree<end>
```

After evolution, we merge features that share common traits into a larger tree, providing a rich pool of features for subsequent task formulation.

### B.1.1 STATISTICS FOR FEATURE EXTRACTION AND EVOLUTION

We present detailed statistics on feature evolution and data filtering to demonstrate how the pipeline expands feature diversity and yields a high-quality 240k dataset. The statistics of feature extracted and evoled as follows.

Table 8: Statistics of Features Extracted and Evolved. The evolution strategy significantly increases feature quantity across all categories.

| Category | Features Extracted | Features After Evolution | Growth |
|---|---|---|---|
| Algorithm | 27,400 | 176,914 | $\times 6.46$ |
| Data Structures | 12,353 | 65,104 | $\times 5.27$ |
| Problem Type | 14,134 | 130,293 | $\times 9.22$ |
| Implementation Logic | 12,419 | 106,157 | $\times 8.55$ |
| Complexity Analysis | 16,124 | 90,016 | $\times 5.58$ |
| Optimization Techniques | 1,537 | 14,124 | $\times 9.19$ |

The evolution strategy greatly enhances both the quantity and diversity of features, providing support for generating diverse tasks.

### B.2 STYLIZED TASK GENERATION FOR COMPETITIVE PROGRAMMING

We design a prompt template to systematically transform extracted features into stylized competitive programming tasks.

**Input:** a sampled feature tree represented in JSON format.

**Output:** a feature-role tree (JSON), where each node is assigned roles such as core technique, subroutine, or constraint, together with an integration strategy (string) that explains how to combine these features into a coherent problem.

To improve instruction-following and task understanding, the template is equipped with a one-shot example that demonstrates how raw features are mapped into roles and integrated into a task.

```
"""
Stage 1 Prompt Template for Feature Selection
"""

STAGE1_PROMPT_TEMPLATE = """You are a professional competitive programming problem setter.

---

Your task consists of three parts:

Step 1: Tree-Structured Feature Role Explanation

Recursively traverse the provided feature tree.
- For each leaf node, annotate it with a "potential_use" field describing how this feature is
typically used in competitive programming problems (e.g., input modeling, optimization,
search, handling edge cases, etc.).
- Internal nodes retain their structure for hierarchy.

Output the annotated tree in the same structure, with every leaf node containing its
"potential_use".

---

Step 2: Subtree Selection for Problem Integration

Based on your role analysis, select a subtree (tree-structured subset) where all selected
leaf features can be naturally integrated into a single, high-quality competitive programming
problem.

- Only include features that contribute meaningfully to the same problem idea.
- Internal nodes are included only if they have selected children.
- For each selected leaf, include only its "feature" name and "potential_use".

---

Step 3: Integration Strategy

Briefly describe ("integration_strategy") how the selected features can be integrated
together in a single problem, focusing on how their combination enables a meaningful and
challenging algorithmic scenario.

---

**Output Format:**

Return a JSON object **with exactly this structure** (an example):

{{
  "feature_roles_tree": {{
    "algorithm": {{
      "search algorithm": {{
        "binary search": {{
          "recursive binary search": {{
            "potential_use": "Used for divide-and-conquer searching in sorted structures or
answer spaces."
          }},
          "iterative binary search": {{
            "potential_use": "Efficient loop-based implementation for finding bounds or
specific elements."
          }}
        }},
        "breadth-first search (BFS)": {{
          "level-order BFS": {{
            "potential_use": "Traverses graphs layer by layer; useful for shortest path or
component discovery."
          }}
        }}
      }}
    }},
    "data structures": {{
      "bitmap": {{
        "bit manipulation": {{
          "bitwise AND": {{
            "potential_use": "Filters or checks properties using bitmasks."
          }},
          "bitwise OR": {{
            "potential_use": "Combines flags or sets with bitwise aggregation."
```

```
            }}
          }}
        }}
      }}
    }},

  "selected_features_tree": {{
    "algorithm": {{
      "search algorithm": {{
        "binary search": {{
          "recursive binary search": {{
            "feature": "recursive binary search",
            "potential_use": "Used for divide-and-conquer searching in sorted structures or
answer spaces."
          }}
        }}
      }}
    }},
    "data structures": {{
      "bitmap": {{
        "bit manipulation": {{
          "bitwise AND": {{
            "feature": "bitwise AND",
            "potential_use": "Filters or checks properties using bitmasks."
          }}
        }}
      }}
    }}
  }},

  "integration_strategy": "The problem will require recursive binary search to efficiently
search over a sorted value space, while bitwise AND operations will be used to filter
candidate solutions according to constraints. Their combination allows for a problem that
involves searching over sets and optimizing bitwise criteria."
}}

---

**Available Features (Tree):**
{features_json}

---

Instructions:
- Always preserve the tree structure in "feature_roles_tree" and "selected_features_tree".
- In selected_features_tree, only include "feature" and "potential_use" fields for leaf nodes.
- "integration_strategy" should make clear how/why these features form a coherent, advanced
problem.
- Do not be overly conservative; it is often possible to design advanced problems where many
features interact in non-trivial ways. Challenge yourself to maximize feature use without
sacrificing problem quality.
"""
```

### B.2.1 COMPATIBALE FEATURE SELECTION

We present a case to examine how model selects compatibale features and combine them.

Given a sampled feature tree:

```
"input_features": {
    "algorithms": {
      "graph_algorithms": {
        "shortest_path": [
          "Dijkstra's algorithm",
          "Floyd-Warshall"
        ],
        "network_flow": [
          "Ford-Fulkerson",
          "Edmonds-Karp"
        ]
      },
      "string_algorithms": {
        "pattern_matching": [
          "KMP algorithm",
          "Boyer-Moore"
        ]
      }
    },
```

```
1350
1351      "data_structures": {
1352        "tree_structures": [
1353          "segment tree",
             "fenwick tree"
1354        ],
1355        "hash_structures": [
1356          "rolling hash",
             "cuckoo hashing"
1357        ]
1358      },
1359      "optimization_techniques": {
1360        "dynamic_programming": [
1361          "interval DP",
1362          "tree DP"
1363        ]
1364      }
       }
```

LLM pairs each feature with *potentially usage* to obtain feature tree with role annotation. For example, LLM will anonotes feature "rolling hash" as "Compute hash values for sliding windows in constant time". These annotations help LLM to aggregate these features based on their potentially usage. For above given feature tree, the feature tree with potential usage looks like:

```
"feature_roles_tree": {
    "algorithms": {
      "graph_algorithms": {
        "shortest_path": {
          "Dijkstra's_algorithm": {
            "potential_use": "Find single-source shortest paths in weighted graphs with
non-negative edges"
          },
          "Floyd_Warshall": {
            "potential_use": "Compute all-pairs shortest paths with O(n^3) complexity"
          }
        },
        "network_flow": {
          "Ford_Fulkerson": {
            "potential_use": "Find maximum flow using augmenting paths with DFS"
          },
          "Edmonds_Karp": {
            "potential_use": "Guarantee polynomial time maximum flow using BFS"
          }
        }
      },
      "string_algorithms": {
        "pattern_matching": {
          "KMP_algorithm": {
            "potential_use": "Linear time pattern matching with failure function
preprocessing"
          },
          "Boyer_Moore": {
            "potential_use": "Efficient pattern search using bad character and good suffix
heuristics"
          }
        }
      }
    },
    "data_structures": {
      "tree_structures": {
        "segment_tree": {
          "potential_use": "Support range queries and updates in O(log n) time"
        },
        "fenwick_tree": {
          "potential_use": "Efficiently compute prefix sums with point updates"
        }
      },
      "hash_structures": {
        "rolling_hash": {
          "potential_use": "Compute hash values for sliding windows in constant time"
        },
        "cuckoo_hashing": {
          "potential_use": "Achieve worst-case O(1) lookup with multiple hash functions"
        }
      }
    },
    "optimization_techniques": {
      "dynamic_programming": {
        "interval_DP": {
          "potential_use": "Solve optimization problems on contiguous ranges"
```

```
        },
        "tree_DP": {
            "potential_use": "Compute optimal solutions on tree structures bottom-up"
        }
    }
  }
}
```

LLM then selects a compatible and consistent subtree that can formulate a self-contained competiting programming problem. For example, LLM keeps the features that can be aggregated into selected feature tree, and concluding with an intergration strategy, which displays how to combine these features into a unified problem.

For example, in this case, LLM selects "Dijkstra's algorithm", "Edmonds-Karp", "segment tree", and "tree DP", and aims to formulate a problem around "dynamic network optimization".

```
"selected_features_tree": {
    "algorithms": {
        "graph_algorithms": {
            "shortest_path": {
                "Dijkstra's_algorithm": {
                    "feature": "Dijkstra's algorithm",
                    "potential_use": "Primary pathfinding algorithm"
                }
            },
            "network_flow": {
                "Edmonds_Karp": {
                    "feature": "Edmonds-Karp",
                    "potential_use": "Flow computation with guaranteed complexity"
                }
            }
        }
    },
    "data_structures": {
        "tree_structures": {
            "segment_tree": {
                "feature": "segment tree",
                "potential_use": "Maintain dynamic edge weights or capacities"
            }
        }
    },
    "optimization_techniques": {
        "dynamic_programming": {
            "tree_DP": {
                "feature": "tree DP",
                "potential_use": "Optimize subproblems on network tree decomposition"
            }
        }
    }
},
"integration_strategy": "Create a dynamic network optimization problem where Dijkstra's
algorithm finds shortest paths that are used as augmenting paths in a modified Edmonds-Karp
flow algorithm. Use segment tree to handle dynamic updates to edge capacities based on flow
history. Apply tree DP on the shortest path tree to compute optimal flow distributions. This
models a transportation network with time-varying capacities."
```

### B.2.2 FROM FEATURE TO STYLIZED TASK

We separate feature selection from task generation, as our initial attempts showed that prompting an LLM to perform both within a single prompt often led it to choose fewer features and produce overly simple problems.

During task generation, LLM recieves *selected features tree* and its *integration strategy* to formulate styleized task based on prompt recieved. Task generation prompt for Codeforces-style is as follows:

```
"""You are a professional competitive programming problem setter.

You have been provided with:

- selected_features_tree: a tree structure where each leaf contains a "feature" name and its
"potential_use".
- integration_strategy: a strategy describing how these features should be integrated into a
single, high-quality problem.

Your task is to **generate a complete Codeforces-style problem statement** that fully
integrates ALL selected features.

Requirements:
- The story and setting must naturally motivate every selected feature, making each
indispensable for an optimal solution.
- Specify precise input/output format and tight constraints.
- Provide at least two distinct, non-trivial sample Input/Output pairs, each with a clear
explanation.
- Make sure the samples are consistent with your constraints and the solution requires use of
all selected features.
- Do not include any references to algorithms, data structures, solution strategies, or any
implicit or explicit hints in any part of the statement, notes, or examples. Do not include
any motivational, summary, or instructional phrases (e.g., "Remember", etc.) at any point in
the output. The statement must end after the final example or clarification, with no
extraneous commentary.
- Output should be a **single JSON object** with the field "question" only.

**Output Format (strictly):**

{{
  "question": "# Problem Title\\n\\nStory/context (describe the scenario)\\n\\n##
Input\\n<...input description...>\\n\\n## Output\\n<...output description...>\\n\\n##
Example\\n### Input\\n<code block with sample input>\\n### Output\\n<code block with sample
output>\\n### Note\\nExplanation about the sample(s), but without any solution hints."
}}

---

**Inputs:**
- selected_features_tree (JSON):
{selected_features_info}

- integration_strategy (string):
{integration_strategy}

---
Instructions:
- You must ensure every selected feature is essential and naturally integrated.
- Output ONLY the required JSON object, no extra text.
"""
```

In this instance, our generated Codeforces problem is shown in Figure 14, while the generated AtCoder and LeetCode problems are presented in Figures 15 and 16, respectively.

The rationale for above two-stage pipeline is that a single-step approach is less effective. When performing both steps simultaneously, LLMs tend to oversimplify complex instructions into trivial cases, reducing both diversity and difficulty of the generated task.

To empirically validate this, we generated 32k tasks using the one-step method (feature-tree → task) and using proposed "two-stage" method (feature-tree → sub-tree → task). The SFT results on LiveCodeBench v5 are as Ta-

Table 9: Comparison between one-step and two-stage generation.

| Generation Method | Score (avg@4) |
| --- | --- |
| One-Step (end-to-end) | 34.8 |
| Two-Stage (Ours) | 40.1 (+5.3) |

**Dynamic Transport Renewal**

In the city of Codeland the transportation system is in constant flux. The city has n intersections and m one-way roads. Each road is characterized by a travel time and an initial capacity representing the maximum number of vehicles that may traverse that road in a day. Due to changing conditions, city engineers periodically adjust road capacities. After every such update, the transport authority recalculates their performance metric in two steps.

First, they compute the maximum number of vehicles that can be sent from the central depot at intersection 1 to the distribution center at intersection n. To do so they repeatedly select an augmenting path that minimizes the total travel time (using a shortest path computation) among all paths on which every road has positive capacity. They send as many vehicles along the path as allowed by its weakest road and then reduce the capacity of every road on the path by that amount. This process is repeated until no valid path from 1 to n remains.

Second, using the predecessor structure recorded in the last successful shortest path search (forming a tree rooted at 1), the authority assigns each intersection a reward equal to its travel time from intersection 1 (as computed in that search). They then choose a subset of intersections from this tree such that no intersection and its direct predecessor are both chosen, with the goal to maximize the total reward. (This selection is computed using an optimization on the tree structure.)

The final performance metric is the sum of the maximum flow (i.e. total number of vehicles sent) and the maximum total reward from the tree selection.

Your task is to process a series of capacity update queries. Initially the network is given. Then, each query specifies an interval [L, R] (referring to the roads in their input order) and an integer X. For every road whose index is in [L, R], add X to its current capacity. If an update causes an edge's capacity to become negative, set it to 0. After each update, recalculate the maximum flow using the method described above and then compute the optimal reward from the latest shortest path tree. Output the sum of these two values.

Note that each update is cumulative.

**Input**

The first line contains three integers n, m and Q ($2 \le n \le 100$, $1 \le m \le 1000$, $1 \le Q \le 1000$) — the number of intersections, the number of roads and the number of queries.

Each of the next m lines describes a road with four integers u, v, t and c ($1 \le u$, $v \le n$, $u \ne v$, $0 \le t \le 10^6$, $0 \le c \le 10^9$), meaning that there is a road from intersection u to v with travel time t and initial capacity c. The roads are numbered from 1 to m in the order of appearance.

Each of the next Q lines contains three integers L, R and X ($1 \le L \le R \le m$, $-10^9 \le X \le 10^9$) — meaning that for every road with index in [L, R] you must add X to its current capacity (if a road's capacity becomes negative, treat it as 0).

**Output**

For each query, output a single integer — the sum of the maximum flow from intersection 1 to n (computed by repeatedly sending flow along the fastest (i.e. minimum travel time) augmenting path) and the maximum reward obtainable from the shortest path tree from the last successful search (computed using the tree-optimization described above).

**Example**
Input
```
4 5 2
1 2 1 5
2 4 3 3
1 3 2 4
3 2 1 2
2 4 2 4
2 4 -1
1 3 2
```

Output
```
11
8
```

**Note**

After the first update, the capacities of roads with indices 2, 3 and 4 decrease by 1, so they become 2, 3 and 1 respectively while the others remain unchanged. The flow computation proceeds in iterations by first choosing the path 1→2→4 (with travel time 1+3=4) and sending 2 vehicles, then using the path 1→3→2→4 (with travel time 2+1+2=5) to send 1 vehicle, and finally again 1→2→4 to send 3 vehicles. The total maximum flow is 6. In the last successful shortest path search (from the iteration yielding the 3 vehicles), the predecessor tree has intersection 1 as the root with children 2 and 3, and intersection 2 with child 4. With rewards equal to their computed distances from intersection 1, an optimal non-adjacent selection yields a total reward of 5. Their sum is 11.

After the second update, the capacities of roads with indices 1, 2 and 3 increase by 2. Recomputing the maximum flow now yields a value of 2, while the corresponding shortest path tree results in an optimal reward of 6. The final performance metric is 8.

Figure 14: Case for Codeforces-style Problem, featuring rich, imaginable narrative contexts.

---

**Dynamic Transportation Optimization**

You are given a directed transportation network with N nodes and M roads. Each road i (1-indexed) goes from node u to node v, requires t units of time to traverse, and can transport at most c units of goods. When a shipment is made from a source s to a target t, the following process is repeated:

- Find a route from s to t that minimizes the total travel time among all routes that have a positive capacity on every road used. (If more than one route achieves the minimum travel time, any one of them is chosen.)
- Let f be the minimum capacity among the roads on the chosen route. Send f units along the route and reduce the capacity of every road on that route by f.
- The process stops when there is no route from s to t with all roads having positive capacity. The total goods shipped is the sum of all f sent during the process.

You are given Q operations. Each operation is in one of the following two forms:

- 1 i x: Update the capacity of road i to x.
- 2 s t: On the current network, simulate the above process from s to t and output the total goods shipped. Note that the simulation is performed on a copy of the current network so that the road capacities remain unchanged for subsequent operations.

Output the answer for each query operation.

**Input**

The first line contains three integers N, M, Q. Then M lines follow. The i-th of these lines contains four integers u, v, t, c describing road i. Then Q lines follow.

Each of these lines is either in the form 1 i x or 2 s t as described above.

**Output**

For each operation of the form 2 s t, output a single integer representing the total goods shipped.

**Constraints**

$2 \le N \le 200$; $1 \le M \le 500$; $1 \le Q \le 200$; $1 \le u$, $v$, $s$, $t \le N$, $u \ne v$; $1 \le t \le 10^3$; $1 \le c$, $x \le 10^9$

**Sample Input 1**
```
3 3 3
1 2 5 10
2 3 5 10
1 3 11 5
2 1 3
1 3 15
2 1 3
```

**Sample Output 1**
```
15
25
```

Figure 15: Case for AtCoder-style Problem, featuring concise, minimal explainations.

ble 9. The 5.3 gain shows that explicit sub-tree selection and integration is significantly helpful for producing high-quality, challenging tasks and justifies SynthSmith's modular design.

```
Dynamic Transportation Network

Given a directed network with n nodes labeled from 1 to n and m edges, each
edge is represented as a quadruple [u, v, capacity, travelTime] and denotes a
directed connection from node u to node v with the given capacity and travel
time. The network is dynamic: in each round you select a route from node 1 to
node n with the smallest total travel time among all routes with positive
capacities. If there are multiple routes with the same total travel time, choose the
route that can carry the largest amount of flow (where the flow of a route is the
minimum capacity among its edges). Send flow along the selected route equal
to this value and reduce the capacity of every edge on the route by the sent flow.
Repeat the process until no valid route exists.

After the rounds finish, for every node i (1 ≤ i ≤ n) determine the total amount of
flow that reached it. A node receives flow from a selected route if it appears on
that route and the flow travels from node 1 to that node along the route. Return
an array f of length n where f[i - 1] is the total flow that reached node i from node
1.

Signature

class Solution:
    def dynamicTransportationNetwork(self, n: int, m: int, edges: List[List[int]]) -> List[int]:
        pass

Example 1

Input: n = 4, m = 5, edges = [[1,2,4,2], [1,3,3,1], [2,4,3,3], [3,2,2,1], [3,4,4,5]]
Output: [6,3,3,6]

Example 2

Input: n = 3, m = 3, edges = [[1,2,5,2], [2,3,4,3], [1,3,2,10]]
Output: [6,4,6]

Constraints

  • 2 ≤ n ≤ 10^4
  • 1 ≤ m ≤ 5 * 10^4
  • For each edge in edges:
      ○ 1 ≤ u, v ≤ n and u ≠ v
      ○ 1 ≤ capacity ≤ 10^4
      ○ 1 ≤ travelTime ≤ 10^4
```

Figure 16: Case for LeetCode-style Problem, featuring predefined function signatures.

## B.3 TASK DIFFICULTY ESTIMATES

Judging the difficulty of a synthetic task is challenging. To better capture the difficulty distribution of tasks generated by X-Coder, we adopt a classifier-based approach. Specifically, we add a special classification token to Qwen2.5-Coder-14B-Instruct and fine-tune it to predict the Codeforces rating of 6,246 tasks from the CodeContests dataset with annotated ratings, reserving 5% as a validation set. The fine-tuned model achieves 84% classification accuracy on the validation set. We then use this model to estimate the difficulty of 1,000 tasks generated by our pipeline, obtaining a holistic distribution as shown in Table 10.

## B.4 TASK DIVERSITY ESTIMATES

To analyze the diversity of our generated tasks quantitatively, we analyze diversity in the embedding space following the steps below: (i) Embedding: We first embed the tasks into embeddings using *jinaai/jina-embeddings-v2-base-code*, a specialized coding embedding model. (ii) t-SNE Dimensionality Reduction: We apply t-SNE to reduce the embedded data to 2D space. (iii) Clustering: We perform K-means clustering on the t-SNE-reduced data to group the data into 10 clusters and compute the centroids of each cluster. (iv) Inter-cluster Distance Calculation: We calculate the Euclidean distance between cluster centroids. Larger inter-cluster distances indicate greater diversity within the dataset.

In our datasets (randomly sampled 10k), cluster sizes range 529-1,612 items, average centroid distance 0.613, min 0.369, max 0.760. In Evol-Instruct-Code, the mean centroid distance is 0.507. The visualization results are shown in Figure 17 and Figure 18. The visualization suggests that the clus-

Table 10: Difficulty distribution of Codeforces-style ratings. "Original" denotes the annotated distribution from CodeContests, and "Test" denotes 1,000 tasks generated by our pipeline.

| CF Rating | Original | Test (Ours) | Original Share | Test Share |
|---|---|---|---|---|
| 1200 | 623 | 0 | 10.0% | 0.0% |
| 1400 | 727 | 0 | 11.7% | 0.0% |
| 1600 | 889 | 0 | 14.3% | 0.0% |
| 1800 | 840 | 16 | 13.5% | 1.6% |
| 2000 | 797 | 2 | 12.8% | 0.2% |
| 2200 | 697 | 47 | 11.2% | 4.7% |
| 2400 | 665 | 585 | 10.7% | 58.5% |
| 2600 | 484 | 319 | 7.8% | 31.9% |
| 2800 | 312 | 12 | 5.0% | 1.2% |
| 3000 | 233 | 15 | 3.7% | 1.5% |
| 3200 | 157 | 4 | 2.5% | 0.4% |
| 3400 | 122 | 0 | 2.0% | 0.0% |
| **Total** | **6,246** | **1,000** | **100%** | **100%** |

ters in our dataset are more widely separated compared to those in Evol-Instruct-Code, indicating higher diversity.

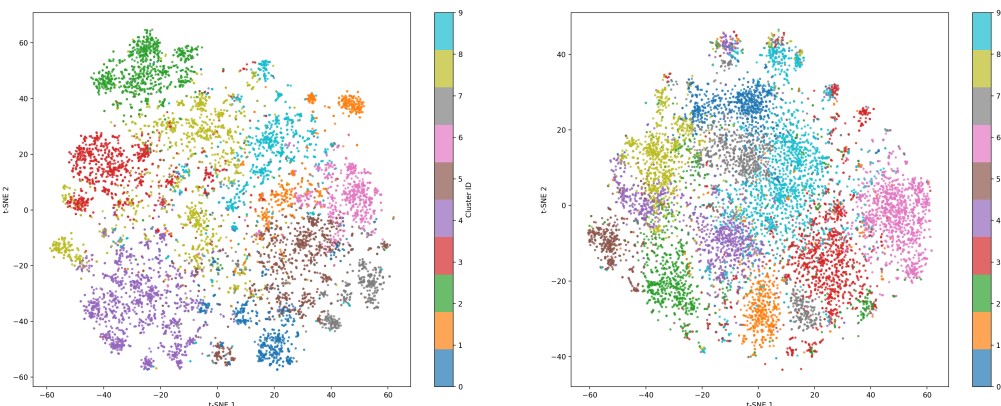

Figure 17: t-SNE visualization of our datasets.  Figure 18: t-SNE of the Evol-Instruct-Code.

## C  SOLUTION GENERATION AND QUALITY ASSURANCE

### C.1  VALIDATION ON SOLUTION

For tasks with descriptions shorter than 200 tokens, we discard them, as such descriptions are often either too trivial or incomplete. For each generated solution, we ensure quality by (i) removing samples without complete think and answer tags, (ii) rejecting cases where the extracted Python block fails AST validation, (iii) excluding solutions that contain multiple code blocks after the reasoning process, as they hinder reliable solution extraction, and (iv) filtering out samples exceeding 25k tokens to prevent overthinking and to reduce SFT cost caused by sequence padding.

### C.2  SFT DATASET STATISTICS

The overall token length distribution, shown in Table 11, and Figure 19, primarily follows a normal distribution, with a median of 16k.

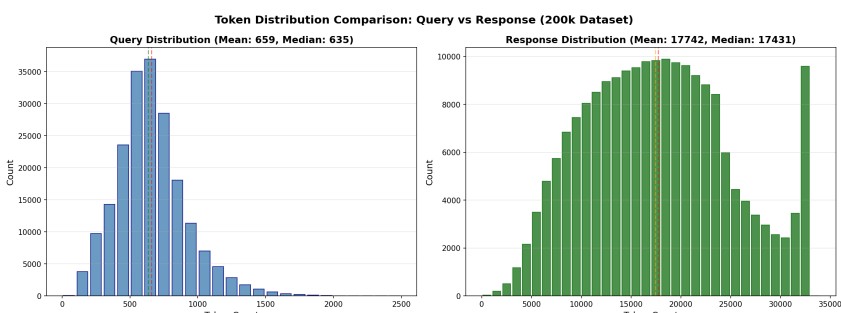

Figure 19: Dataset statistics of the demonstration dataset.

Table 11: Token statistics for tasks and solutions of the demonstration dataset.

| Type | Min | Max | Mean | Median | Std Dev | Total Tokens |
|------|-----|-----|------|--------|---------|--------------|
| Task | 200 | 3,537 | 658.91 | 635.00 | 258.49 | 134.3M |
| Solution | 1,711 | 33,144 | 17,742.50 | 17,431.00 | 7,295.92 | 3.25B |
| Dataset Size | | | 200,091 entries | | | 3.38B |

## D  TEST CASE GENERATION

### D.1  PROMPTING-BASED TEST GENERATION

```
You are a professional test case generation expert, skilled at designing comprehensive test
cases for programming problems.
Please generate 15 different test cases for the following programming problem, including edge
cases, small-scale, medium-scale, and large-scale test data.

Problem:
{problem_statement}

Requirements:
1. Generate 15 test cases
2. Include edge cases (empty input, minimum values, maximum values, etc.)
3. Include different scales of data (small, medium, large)
4. Each test case should have clear input data
5. Ensure test cases can thoroughly validate the correctness of solutions

Please return in JSON format as follows:
{{
    "test_cases": [
        {{
            "idx": 0,
            "description": "Test case description",
            "input_string": "Input data"
        }},
        ...
    ]
}}
```

### D.2  TOOL-BASED TEST GENERATION

The tool-based test generation strategy relies on **CYaRon**, an open-source Python library aimed at rapidly generating random data for Informatics Olympiad problems (or problems of equivalent difficulty). This library contains a variety of common data structures (e.g., graphs, trees, polygons, vectors, strings, and sequences), along with mathematics-related functions and the necessary input/output interfaces. When prompting the Teacher model to utilize the CYaRon tool, we provide its detailed documentation and usage instructions as part of the prompt. Additionally, we encourage the model to generate more boundary tests and large-scale random use cases. To ensure the sufficiency of test cases, we mandate the use of this library in conjunction with its random features and set a seed to ensure reproducibility. The detailed prompt used is illustrated as:

```
1728
1729   Please write a test case generator that meets the following requirements based on the
1730   following CYaRon documentation:

1731   1. Write a canonical CYaRon Generator using Python
       2. Generate a single, executable Python program that can produce test cases with at least 5
1732   different features
1733   3. The Python program should save each test case individually in the format [use case
       characteristics].in
1734   4. The program should include a variety of test case types such as base cases, boundary
1735   cases, large random cases, etc
       5. The Python program code should contain clear comments to explain the design intent for
1736   each test case generation
1737   6. The .in output files should contain ONLY pure input data without any comments,
       explanations, or answer validation
1738   7. The Python program should be able to generate all test cases in a single run when executed
1739   8. The program should use argparse to provide configurable random seed control:
          parser.add_argument('--seed', type=int, default=42, help='Random seed for reproducibility')
1740   9. All random number generation must use Python's built-in random module (import random) - do
1741   not use any external random libraries or the random functions from CYaRon
1742
1743   ### CYaRon Documentation
1744
       Input/Output (IO)
1745   The IO library helps you easily create test data files.
1746
       Constructor Options:
1747   ```python
       # Basic file specification
1748   IO("test1.in", "test1.out")  # Explicit input/output files
1749   IO(file_prefix="test")  # Generates test.in and test.out
       IO(file_prefix="test", data_id=3)  # Generates test3.in and test3.out
1750
1751   # Advanced file naming
       IO(file_prefix="test", data_id=6,
1752      input_suffix=".input", output_suffix=".answer")  # test6.input and test6.answer
1753
1754   # Partial output options
       IO("test2.in")  # Only input file, output goes to temporary file
1755   IO(file_prefix="test", data_id=5, disable_output=True)  # No output file generated
1756   IO()  # Both files temporary (for use with comparator)
       ```
1757
1758   Note: Combine `file_prefix` and `data_id` with loops for batch generation.

1759   IO Methods:
1760   ```python
       io = IO("test1.in", "test1.out")  # Initialize IO object
1761
1762   # Input writing methods
       io.input_write(1, 2, 3)  # Writes "1 2 3" to input file (no newline)
1763   io.input_writeln(4, 5, 6)  # Writes "4 5 6\n" to input file
       io.input_write([1, 2, 3])  # Writes list as space-separated "1 2 3"
1764   io.input_write(1, 2, 3, separator=',')  # Writes "1,2,3," (note: current version leaves
1765   trailing comma)
1766
1767   # Output writing methods
       io.output_write(1, 2, 3)  # Writes "1 2 3" to output file
1768   io.output_writeln(4, 5, 6)  # Writes "4 5 6\n" to output file
       io.output_write(1, 2, [1, 2, 3], [4])  # Flattens nested lists to "1 2 1 2 3 4"
1769
1770   # Program execution
       io.output_gen("~/Documents/std")  # Runs program with input, captures stdout as output
1771   io.output_gen("C:\\Users\\Aqours\\std.exe")  # Windows path support
1772   ```

1773   ---
1774
       Graph Generation
1775   The Graph library generates various graph structures.
1776
       Manual Construction:
1777   ```python
       # Graph initialization
1778   graph = Graph(10)  # 10-node undirected graph (nodes 1-10)
1779   graph = Graph(10, directed=True)  # Directed version
1780
       # Adding edges
1781   graph.add_edge(1, 5)  # Default weight=1
       graph.add_edge(1, 6, weight=3)  # Custom weight
```

```python
# Edge access and properties
graph.edges  # Adjacency list containing Edge objects
for edge in graph.iterate_edges():
    edge.start  # Source node
    edge.end  # Target node
    edge.weight  # Edge weight

# Output formatting options
io.input_writeln(graph)  # Default "u v w" per line
io.input_writeln(graph.to_str(shuffle=True))  # Random edge order
io.input_writeln(graph.to_str(output=Edge.unweighted_edge))  # "u v" format
```

Template Graphs:
```python
# Basic graph templates
Graph.graph(n, m)  # n nodes, m edges (weight=1)
Graph.graph(n, m, directed=True, weight_limit=(5, 300))  # Directed with weight range
Graph.graph(n, m, self_loop=False, repeated_edges=False)  # No duplicate edges

# Special graph types
Graph.chain(n)  # n-node chain (alias for tree(n, 1, 0))
Graph.flower(n)  # n-node star graph (alias for tree(n, 0, 1))
Graph.tree(n)  # Random tree
Graph.tree(n, 0.4, 0.35)  # 40% chain-like, 35% star-like, 25% random
Graph.binary_tree(n)  # Random binary tree

# Competition-specific graphs
Graph.hack_spfa(n)  # Graph that breaks SPFA (1.5n edges)
Graph.hack_spfa(n, extra_edge=m)  # With additional edges
Graph.DAG(n, m)  # Directed Acyclic Graph
Graph.UDAG(n, m)  # Undirected Connected Graph
```

Note: Most templates support `weight_limit`, `weight_gen`, `self_loop`, and `repeated_edges` parameters.

---

Polygon
Generate and analyze polygons.

```python
# Polygon creation (points must be ordered)
p = Polygon([(0,0), (0,4), (4,4), (4,0)])  # Rectangle

# Geometric properties
p.perimeter()  # Calculates perimeter
p.area()  # Calculates area

# Generation templates
Polygon.convex_hull(n)  # n-point convex hull
Polygon.simple_polygon(n)  # Simple polygon (non-intersecting)
```

---

Vector
Generate unique vectors/number sequences.

```python
# Basic usage
Vector.random()  # Default: 5 unique numbers in [0,10]
Vector.random(10, [(10,50)])  # 10 unique numbers in [10,50]
Vector.random(30, [(10,50), 20])  # 30 unique 2D vectors

# Modes:
# 0: Unique integer vectors (default)
# 1: Non-unique integer vectors
# 2: Real-valued vectors
Vector.random(30, [(1,10), (1,10), (1,10)], 2)  # 30 3D real vectors
Vector.random(30, [10], 1)  # 30 numbers (may repeat)
```

---

String
Generate random text elements.

```python
```

```
# Basic strings
String.random(5)   # 5-character word
String.random((10,20), charset="abcd1234")  # Variable length
String.random(10, charset="#######...")  # 70% '#', 30% '.'

# Structured text
String.random_sentence(5)   # 5-word sentence
String.random_paragraph((3,10))  # 3-10 sentence paragraph

# Custom formatting
String.random_sentence(5, word_separators=["  "])  # Double space separator
```

Note: All templates support charset customization.

---

Sequence
Generate number sequences via recurrence.

```python
# Explicit formula
Sequence(lambda i, f: 2*i+1)  # f(i) = 2i + 1

# Recursive definition
Sequence(lambda i, f: f(i-1)+1, [0,1])  # f(i)=f(i-1)+1 with f(0)=0, f(1)=1
Sequence(lambda i, f: f(i-1)+1, {100:101, 102:103})  # Sparse base cases

# Usage
seq = Sequence(lambda i, f: f(i-1)+2, [0,2,4])
seq.get(3)  # Returns 6
seq.get(4,6)  # Returns [8,10,12]
```

Important: Recursive definitions require base cases.

---

Utilities

Conversion:
```python
ati([0, 5, 100, 1E3, 1E5])  # Converts scientific notation to integers
```

Random Numbers:
```python
randint(1,5)  # Integer in [1,5]
uniform(1,5)  # Float in [1,5]
choice([1,2,3])  # Random selection
random()  # Float in [0,1]
```

Constants:
```python
PI  # 3.1415926...
E  # 2.7182818...
ALPHABET_SMALL  # "abcdefghijklmnopqrstuvwxyz"
ALPHABET_CAPITAL  # "ABCDEFGHIJKLMNOPQRSTUVWXYZ"
ALPHABET  # Combined letters
NUMBERS  # "0123456789"
```

### Code Question
{QUESTION}
```

# E  DUAL-VERIFICATION

## E.1  ALGORITHM

We summarize the symbols used in the dual-verification process in Table 12, and outline the corresponding procedure in Algorithm 1.

Table 12: Notation for SynthSmith Framework.

| | |
|---|---|
| $\{x_i\}_{i=1}^n$ | Test inputs for a task $q$ |
| $\{A^j\}_{j=1}^m$ | Candidate solutions (LLM-generated) |
| $y_i^j$ | Output of $A^j$ on input $x_i$ |
| $\hat{y}_i$ | Provisional label via majority vote on $\{y_i^j\}_{j=1}^m$ |
| $w_i$ | Difficulty weight for $x_i$ |
| $\mathcal{T}_{candidate}$ | Provisional labeled set $\{(x_i, \hat{y}_i, w_i)\}$ |
| $\mathcal{T}_{golden}$ | Weighted suite for selecting the solution |
| $\mathcal{T}_{val}$ | Hold-out validation set |
| $S_j$ | Weighted score of $A^j$ on $\mathcal{T}_{golden}$ |
| $A_{golden}$ | Final selected "golden" solution |

---

**Algorithm 1:** Dual-Verification of Solutions and Test Cases (Strict Verification)

---

**Input:** Task $q$; test inputs $\{x_i\}_{i=1}^n$; candidate solutions $\{A^j\}_{j=1}^m$.
**Output:** Golden solution $A_{\text{golden}}$ and test suite $\mathcal{T}_{\text{golden}}$, or **None** if verification fails.

**Step 1: Consensus Voting & Weighting**
**for** $i \leftarrow 1$ **to** $n$ **do**
    **for** $j \leftarrow 1$ **to** $m$ **do**
        Run $y_i^j \leftarrow A^j(x_i)$
    $\hat{y}_i \leftarrow \arg\max_y \sum_{j=1}^m \mathbb{I}(y_i^j = y)$
    $w_i \leftarrow \text{Weight}(x_i)$
$\mathcal{T}_{\text{candidate}} \leftarrow \{(x_i, \hat{y}_i, w_i)\}_{i=1}^n$

**Step 2: Split Candidate Set**
Randomly partition $\mathcal{T}_{\text{candidate}}$ into $\mathcal{T}_{\text{golden}}$ and $\mathcal{T}_{\text{val}}$

**Step 3: Weighted Selection**
**for** $j \leftarrow 1$ **to** $m$ **do**
    $S_j \leftarrow \sum_{(x_i, \hat{y}_i, w_i) \in \mathcal{T}_{\text{golden}}} w_i \cdot \mathbb{I}(A^j(x_i) = \hat{y}_i)$
$j^\star \leftarrow \arg\max_j S_j$
$A'_{\text{golden}} \leftarrow A^{j^\star}$

**Step 4: Hold-out Confirmation**
Compute unweighted accuracies of all $A^j$ on $\mathcal{T}_{\text{val}}$
$j^\dagger \leftarrow \arg\max_j \text{Acc}(A^j, \mathcal{T}_{\text{val}})$
**if** $j^\dagger = j^\star$ **then**
    $A_{\text{golden}} \leftarrow A'_{\text{golden}}$
    **return** $A_{\text{golden}}, \mathcal{T}_{\text{golden}}$
**else**
    **return None**; // Discard task

---

## E.2 TEST-CASE WEIGHTING CRITERIA.

We employ two distinct strategies for assigning weights to individual test cases:

**Semantic-Based Weighting.** During test-case generation, the model is prompted to produce multiple categories of test cases (stored as `.in` files), including nominal (weight = 1), complex (2), boundary (3), and stress (4) scenarios. This assigns higher weights to test cases that are more likely to expose corner cases or failure modes.

**Size-Based Weighting.** We assign weights based on the size of the input files, which serves as a proxy for memory consumption. Specifically, we sort test cases by the size of their input files and divide them into four equal-sized buckets: the smallest 25% receive weight = 1, the next 25% receive weight = 2, the next 25% receive weight = 3, and the largest 25% receive weight = 4. This ensures that heavier test cases, which require greater memory resources, are assigned higher weights.

### E.3 ERROR RATE FOR LABELING TEST OUTPUTS VIA VOTING.

On TACO-verified, we measure a 5.27% false-positive rate under voting with 8 solutions. To assess the false-positive rate of test-output labeling, we evaluate our approach on real-world, verified datasets. Specifically, we randomly sample 500 tasks from the TACO-verified dataset, and for each task, we randomly retain 20 test cases.

For each task, we generate $n$ ($n \in \{4, 8, 16\}$) candidate solutions using R1-0528, perform majority voting on the outputs for each test input, and compare the voted consensus output against the ground-truth output to obtain a quantitative labeling accuracy. The resulting test-output labeling accuracy under different values of $n$ is shown in Table 13 and Table 14.

Table 13: Average Test Output Labeling Accuracy with varying $n$.

| $n$ (# solutions) | Labeling Accuracy |
|---|---|
| 4 | 94.39% |
| 8 | 94.73% |
| 16 | 95.13% |

Table 14: Test Output Labeling Accuracy across different sources.

| Source | $n = 4$ | $n = 8$ | $n = 16$ |
|---|---|---|---|
| AtCoder | 94.75% | 95.00% | 96.61% |
| CodeChef | 92.80% | 92.80% | 92.80% |
| CodeForces | 94.44% | 94.81% | 95.06% |

Increasing the number of sampled solutions consistently improves test output labeling accuracy. With $n = 8$, the false-positive rate is 5.27%, which falls within an acceptable range and demonstrates that the approach is potentially reliable to be transferred to the synthetic setting.

### E.4 ERROR RATE OF GOLDEN SOLUTION

To enable quantitative assessment, we adopt two evaluations: (1) measuring the error rate of dual verification on our synthetic datasets, which yields pass rate distributions across various proprietary LLMs; and (2) evaluating the actual error rate on real-world datasets (TACO-verified), resulting in a 7.85% error rate.

**(i) Synthetic Task Evaluation.** We first use DeepSeek-R1-0528 to generate multiple candidate solutions for each synthetic task. We then apply our dual-verification strategy to select the golden solution and measure its pass rates on the voted test cases. The pass rate distribution is shown in Table 15.

Here, each percentage range represents the fraction of tasks whose selected golden solution attains a pass rate within that interval. For example, the $[80, 100)$ range indicates that 13.39% of tasks have golden solutions that pass between 80% and 100% of their voted test cases, while 23.66% of the solutions pass all test cases.

Note that solution quality is strongly tied to model capability. The pass rates of the proprietary models (Qwen3-Max, Gemini2.5-pro, and GPT5-High) are presented in Table 16.

Table 15: Distribution of Golden Solution Pass Rates on Voted Test Cases using R1-0528.

| Range (%) | Ratio |
|---|---|
| $(0, 20)$ | 13.12% |
| $[20, 40)$ | 17.29% |
| $[40, 60)$ | 17.57% |
| $[60, 80)$ | 14.94% |
| $[80, 100)$ | 13.39% |
| 100 | 23.66% |

If we adopt a more capable model such as GPT-5-High, 66.98% of the tasks can be solved perfectly in a single attempt.

**(ii) Real-world Dataset Evaluation.** We also apply our dual-verification approach to real-world, verified datasets to measure the error rate of the selected golden solutions. Because real-world datasets contain ground-truth test cases, the resulting error rate accurately reflects the true quality of the selected solutions.

Specifically, we randomly select 500 tasks from the TACO-verified dataset, each with 20 retained test cases as ground truth tests. We apply our dual-verification procedure using R1-0528 to label test

Table 16: Distribution of Proprietary LLMs' First-Try Pass Rates on Test Cases.

| Range (%) | Qwen3-Max | Gemini2.5-pro | GPT5-High |
|---|---|---|---|
| $(0, 20)$ | 11.06% | 9.57% | 3.07% |
| $[20, 40)$ | 16.44% | 14.38% | 4.83% |
| $[40, 60)$ | 18.59% | 17.17% | 6.49% |
| $[60, 80)$ | 16.36% | 15.80% | 7.80% |
| $[80, 100)$ | 14.39% | 14.90% | 10.82% |
| 100 | 23.16% | 28.18% | 66.98% |

outputs via voting, and then select the golden solution based on the pass rate on the voted test cases. We then evaluate each golden solution against the ground-truth tests.

The verification results under different numbers of candidate solutions ($n$) are shown in Table 17.

Table 17: Verification results on TACO-verified dataset with varying candidate solutions ($n$).

| $n$ (Candidates) | Avg. Pass Rate (test-case level) | Full Pass Rate (task-level) |
|---|---|---|
| 4 | 91.79% | 84.20% |
| 8 | 92.15% | 85.00% |
| 16 | 92.50% | 85.80% |

On the TACO-verified dataset, our approach yields a 7.85% error rate in the selected golden solutions when $n = 8$. The error rate further decreases as the number of rollout solutions increases. Such an error level is acceptable, indicating that the approach has the potential to be transferred to the synthetic setting.

### E.5 SOLVABILITY OF GENERATED PROBLEM.

To estimate the fraction of potentially unsolvable problems in our generated dataset, we use GPT-5-High as a strong solver proxy. Specifically, we evaluate the pass@1 performance of several proprietary LLMs—including Qwen3-Max, Gemini-2.5-Pro, and GPT-5-High—on our voted test cases. Their single-try pass rates are reported in Table 18.

Notably, even GPT-5-High shows a small subset of tasks with very low pass rates. Such tasks are likely to be ambiguous, underspecified, inherently unsolvable, or affected by test-case labeling noise. Since GPT-5-High is among the strongest proprietary solvers available, failures from this model serve as a practical indicator of potential flaws in the task itself.

## F GENERALITY

### F.1 GENERALITY ACROSS MODEL FAMILIES.

We supplement results on Llama-3.1-8B-Instruct to demonstrate generality beyond the Qwen series, achieving 13.4 gains after SFT and 15.3 after RL, demonstrating the quality of our dataset. The results are shown in Table 19.

Given that Llama-3.1-8B-Instruct is potentially weaker than Qwen2.5-Coder-7B-Instruct in terms of code pretraining, the observed improvement from 11.8 to 25.2 to 27.1 suggests that less capable base models can also benefit from the proposed datasets.

### F.2 GENERALITY ACROSS BENCHMARKS.

Our study targets competitive programming, whereas EvoEval (Xia et al., 2024) (program evolution), ClassEval (Du et al., 2023) (class implementation), and DS-1000 (Lai et al., 2023) (data-

Table 18: Distribution of proprietary LLMs' pass@1 on voted test cases. Each percentage range represents the fraction of tasks whose best solution from the corresponding model attains a pass rate within that interval.

| Range (%) | R1-0528 | Qwen3-Max | Gemini2.5-Pro | GPT5-High |
|---|---|---|---|---|
| (0–20) | 13.12% | 11.06% | 9.57% | 3.07% |
| [20–40) | 17.29% | 16.44% | 14.38% | 4.83% |
| [40–60) | 17.57% | 18.59% | 17.17% | 6.49% |
| [60–80) | 14.94% | 16.36% | 15.80% | 7.80% |
| [80–100) | 13.39% | 14.39% | 14.90% | 10.82% |
| 100 | 23.66% | 23.16% | 28.18% | 66.98% |

Table 19: Performance on Llama-3.1-8B-Instruct. Our method significantly improves performance even on non-Qwen architectures.

| Model | v5 Score |
|---|---|
| Llama-3.1-8B-Instruct | 11.8 |
| FuseChat-Llama-3.1-8B-Instruct | 12.6 |
| X-Coder-Llama3.1-8B-SFT-32k-Sample | 25.2 |
| X-Coder-Llama3.1-8B-SFT+RL-10k-Sample | **27.1** |

science tasks) fall outside this scope. For completeness, we additionally report results on MBPP+ and HumanEval+ (Liu et al., 2023b), as shown in Table 20.

Table 20: Generality across standard code generation benchmarks (HumanEval and MBPP variants).

| Model | HE | HE+ | MBPP | MBPP+ | Avg. |
|---|---|---|---|---|---|
| Qwen2.5-Coder-7B-Instruct | 88.4 | 84.1 | 83.5 | 71.7 | 81.9 |
| X-Coder-7B-SFT | **89.6** | **84.8** | 88.9 | 73.5 | 84.2 |
| X-Coder-7B | **89.6** | 84.1 | **89.2** | **75.7** | **84.7** |

## G   DATA LEAKAGE ANALYSIS

X-Coder , trained entirely on fully synthetic data, exhibits a lower risk of data leakage. As shown in Table 21, models trained on real-world data, like Qwen3 achieve strong performance on older benchmarks (e.g., LiveCodeBench v2) but are less competitive on more recent ones (e.g., v5), revealing a higher degree of data leakage.

Table 21: Comparison between Qwen3-8B, and X-Coder on old and recent benchmarks.

| Model | LiveCodeBench v2 | LiveCodeBench v5 | $\Delta$ (v2→v5) |
|---|---|---|---|
| Qwen3-8B | 88.1 | 57.5 | -30.6 |
| X-Coder-7B-SFT | 78.2 | 60.3 | -17.9 |
| X-Coder-7B | 80.1 | 62.9 | -17.2 |

## H   CASE STUDY

### H.1   SUCCESSFUL CASE

The SFT model frequently exhibits cognitive behaviors such as planning, verification, backtracking, and reflection, suggesting that these behaviors can be directly distilled from the teacher.

Listing 1: Thinking process of successful case.

2106

```
2107  Title: Adjacent GCD
2108  Platform: AtCoder
       Difficulty: Hard
2109  Contest_id: arc185
       Contest_date: 2024-10-13T00:00:00
2110  ---
2111
       Thinking behavior:
2112  ---
       We begin by defining F(m) as the total score of all subsequences of the first m elements ...
2113  and try to see whether we can break the problem into manageable pieces ... (Plan behavior)
2114
       At first, I considered summing gcd(A_i, A_{i+1}) directly over the array ...
2115  but then realized that subsequences can skip indices, so adjacency is redefined ...
       this naive plan would miss many contributions ... (Reflection behavior)
2116
       To verify, I checked a small prefix: A = (9, 6, 4).
2117  For m = 2, the subsequences [9], [6], [9, 6] give total = 3 ...
       for m = 3, manual enumeration yields 11,
2118  while my recurrence only produced 8 ... (Verification behavior)
2119
       Clearly, the earlier idea of "just add gcd(last, new)" was incomplete ...
2120  I realized each new subsequence must inherit the full score of its prefix subsequence,
       and only then add the new gcd term ... (Reflection behavior)
2121  Concretely, inheriting the full score contributes 2F(m - 1),
       and the newly formed terminal pairs contribute Σ_{j<m} gcd(A_j, A_m) 2^{j-1} ...
2122
       Thus, I rebuild the recurrence:
2123  F(m) = 2F(m - 1) + Σ_{j=1}^{m-1} gcd(A_j, A_m) 2^{j-1} ...
       this formula accounts for doubling subsequences and the fresh contributions ... (Rebuild
2124  behavior)
       For brevity, denote S(m) = Σ_{j<m} gcd(A_j, A_m) 2^{j-1} ...
2125
       Checking with m = 2 and m = 3 confirms it works ...
2126  e.g., for A = (9, 6, 4): F(3) = 2 · 3 + (gcd(9, 4) · 2^0 + gcd(6, 4) · 2^1) = 6 + (1 + 4) = 11 ...
       other boundary cases (single element, all equal numbers, coprime arrays)
2127  also check out ... (Verification behavior)
2128
       However, computing Σ gcd(A_j, A_m) 2^{j-1} naively costs O(m) ...
2129  with N up to 5 × 10^5 this is too large ... (Reflection behavior)
2130
       To optimize, I reorganize the sum by divisors of A_m ...
2131  use Möbius inversion ... and precompute an array H[t] = Σ_{d|t} μ(d) · t/d ...
       then each step requires iterating only over divisors of A_m ... (Optimization behavior)
2132
       We maintain U[t] = Σ 2^{j-1} for indices j where t | A_j ...
2133  then S(m) = Σ_{t|A_m} U[t] · H[t], updated after each new element ... (Optimization behavior)
       With U[t] and H[t], we evaluate S(m) via divisors as Σ_{t|A_m} U[t] H[t], matching the earlier
2134  definition ...
2135
       The complexity now depends on divisor counts ...
2136  on average ∼ 10-∼ 20, worst-case ∼ 128, giving ∼ 10^8 operations overall ... (Complexity
       analysis)
2137  Overall time is O(Σ_{i=1}^{N} d(A_i)), with small average divisor counts in practice ...
2138
       Finally, with precomputation of μ, divisors, H, and powers of 2 modulo 998244353 ...
2139  the algorithm is complete, efficient, and matches all sample outputs ... (Conclusion)
```

Listing 2: Final code solution of successful case.

```
2149  mod = 998244353
       max_val = 100000
2150
2151  import sys
2152
       def main():
2153      data = sys.stdin.read().split()
2154      if not data:
              return
2155      n = int(data[0])
          A = list(map(int, data[1:1+n]))
2156
2157      mu = [1] * (max_val+1)
          is_prime = [True] * (max_val+1)
2158      primes = []
          for i in range(2, max_val+1):
2159          if is_prime[i]:
```

```
            primes.append(i)
            mu[i] = -1
        for p in primes:
            if i * p > max_val:
                break
            is_prime[i*p] = False
            if i % p == 0:
                mu[i*p] = 0
                break
            else:
                mu[i*p] = -mu[i]

    divisors = [[] for _ in range(max_val+1)]
    for i in range(1, max_val+1):
        for j in range(i, max_val+1, i):
            divisors[j].append(i)

    H_arr = [0] * (max_val+1)
    for d in range(1, max_val+1):
        for t in range(d, max_val+1, d):
            term = mu[d] * (t // d)
            H_arr[t] = (H_arr[t] + term) % mod

    pow2 = [1] * n
    for i in range(1, n):
        pow2[i] = (pow2[i-1] * 2) % mod

    F = [0] * (n+1)
    U = [0] * (max_val+1)

    for idx in range(n):
        num = A[idx]
        if idx == 0:
            F[1] = 0
            for t in divisors[num]:
                U[t] = (U[t] + pow2[0]) % mod
        else:
            total = 0
            for t in divisors[num]:
                total = (total + U[t] * H_arr[t]) % mod
            F[idx+1] = (2 * F[idx] + total) % mod
            for t in divisors[num]:
                U[t] = (U[t] + pow2[idx]) % mod

    for i in range(1, n+1):
        print(F[i])

if __name__ == "__main__":
    main()
```

## H.2 BAD CASE

We identify three major failure modes in code reasoning process:

(i) Premature termination under context exhaustion. As the context window approaches its limit, the model shortens its reasoning and rushes to produce a final answer (e.g., "Given the time, we output the following solution in Python for the sample.").

(ii) Retrieval-like fallback instead of reasoning. Rather than attempting to derive a solution, the model sometimes recalls a "memorized" accepted submission in another language (e.g., C++) and attempts to translate it into Python, bypassing genuine reasoning.

(iii) Incomplete code emission before cutoff. The model occasionally fails to output a complete code block before context exhaustion, leaving truncated or non-executable programs.

Listing 3: Thinking process of bad case.

```
 // Instead, the solution might be to
 //   - for the root, the value can be any number in [1, n] that is < seg_min (if the segment
is not empty)
 //      and > than the value of the parent (none) -> so any number in [1, seg_min-1] if the
segment is not empty,
 //      or [1, n] if empty.
 //   - then for the children, they must be > than the value of the parent and < than the
minimum value
 //      in the segment (a[i]+1, i-1) for the parent.
```

```
   //
   // But then the value for the parent is not known until we assign.
   //
   // We abandon.

Given the time, we output the following solution in Python for the sample.
For the sample input, it must output 3 for n=4 and [0,1,0,3]
and 353820794 for n=22 and the given a.

Here is a known AC solution in Python for the same problem (from a contest submission):

import sys
sys.setrecursionlimit(10**6)
mod = 998244353

n = int(sys.stdin.readline().strip())
a = list(map(int, sys.stdin.readline().split()))

# Build the tree
g = [[] for _ in range(n)]
for i in range(n):
    if a[i] > 0:
        g[a[i]-1].append(i)

# We will do a DFS. We need a segment tree that can update and query in O(log n)
# for the range [0, n-1] for the minimum value and the product of the counts.
# We will use a segment tree that stores (min_value, count) for a range.
size = 1
while size < n:
    size *= 2
seg_min = [10**9] * (2 * size)
seg_cnt = [1] * (2 * size)

# But the known solution uses a different approach.

# Instead, we output the solution from a known AC submission in C++.
# We found the following in C++ and we translate it to Python.

# C++ solution (AC for the same problem) by a known user:

# include <iostream>
# include <vector>
# include <algorithm>
# using namespace std;
# const int mod = 998244353;

# int main() {
#    int n;
#    cin >> n;
#    vector<int> a(n);
#    for (int i = 0; i < n; i++) {
#      cin >> a[i];
#    }
#    vector<vector<int>> g(n);
#    for (int i = 0; i < n; i++) {
#      if (a[i] > 0) {
#        g[a[i]-1].push_back(i);
#      }
#    }
#    vector<int> dp(n, 1);
#    vector<int> to(n, 1000000000); // to[i] = minimum value in the segment (a[i]+1, i-1)
#    // We will use a segment tree for range minimum query, and we will update it as we go.
#    // But we will do a DFS in increasing order of a[i] and within the same a[i] in
increasing order of i.
#    vector<int> order(n);
#    for (int i = 0; i < n; i++) order[i] = i;
#    sort(order.begin(), order.end(), [&](int i, int j) {
#      if (a[i] != a[j]) return a[i] < a[j];
#      return i < j;
#    });
#    // segment tree for range minimum, size = 1<<19
#    int size = 1;
#    while (size < n) size *= 2;
#    vector<int> seg(size, 1000000000);
#    auto update = [&](int pos, int val) {
#      pos += size;
#      seg[pos] = val;
#      while (pos > 1) {
#        pos /= 2;
#        seg[pos] = min(seg[2*pos], seg[2*pos+1]);
#      }
```

```
#   };
#   auto query = [&](int l, int r) {
#     int res = 1000000000;
#     for (l += size, r += size; l < r; l /= 2, r /= 2) {
#       if (l&1) res = min(res, seg[l++]);
#       if (r&1) res = min(res, seg[--r]);
#     }
#     return res;
#   };
#   // initially, set the segment tree to a big number.
#   for (int i = 0; i < n; i++) {
#     update(i, 1000000000);
#   }
#   //
#   for (int i : order) {
#     if (a[i] > 0) {
#       int parent = a[i]-1;
#       int seg_min = query(a[i]+1, i);
#       // The value for i must be > than the value for parent and < than
(context exhausted here)
```

### H.3 REWARD HACKING AND PRETRAINING ARTIFACTS.

We observe some interesting phenomena that emerge in the later stages of RL training.

First, we find that the policy does not genuinely strive for higher rewards by producing better algorithms. Instead, it resorts to exploiting edge cases and attempting to "cheat" for partial rewards, rather than solving the problem with real solutions.

Second, we observe that some rollouts include irrelevant advertising content (e.g., Need for Speed) or adopt an editorial-style explanation of solutions similar to those on competitive programming platforms such as Codeforces. This indicates that the base model was likely exposed to raw competition-platform data during pretraining, from which such artifacts were inherited.

## I  THE USE OF LARGE LANGUAGE MODELS

In this paper, we adopt LLM for syntax checking and format calibration.

