# OpenReview forum: "X-Coder: Advancing Competitive Programming with Fully Synthetic Tasks, Solutions, and Tests"
_ICLR.cc/2026/Conference — Submitted to ICLR 2026_

### Official Review · Reviewer_vHC5 · 2025-10-27

**Soundness:** 3
**Presentation:** 3
**Contribution:** 2
**Rating:** 4
**Confidence:** 4

**Summary:**

This paper proposes SynthSmith, a feature-based pipeline to fully synthesize competitive programming dataset including problem statements, solutions, and unit tests. Starting from Qwen2.5-Coder-7B-Instruct and Qwen3-8B-Base, the authors use the synthetic SFT and RL dataset to train the X-Coder series. The manuscript is centering around the synthetic dataset pipeline, covering task (statement) generation, solution generation, and unit tests generation, in which the authors use "dual-verification", a variant based on majority voting with custom weights to select ground truth solutions and unit tests set. The authors follow a standard SFT-then-RL training to show the effectiveness of the synthetic dataset. The paper further analyzes scaling (unique tasks > multiple solutions per task), SFT-then-RL dynamics (“good-gets-better”), and ablations (verification, long- vs short-CoT, task style).

**Strengths:**

The work is well-executed (the various prompt pipelines), and the results look strong. The main novelty of the paper is the synthetic code pipelines, where the authors detail the prompt used in the Appendix to demonstrate how the features (algorithm/data structure) are extracted, grow, and generate a problem statement. Additional experiments are conducted to show the scaling effect of the dataset. Some extra ablations are also conducted, such as the tool-based generation vs prompt-based generation for unit tests. I also appreciate the authors' commitment to reproducibility (while I leave some doubts on it with details later).

**Weaknesses:**

Since the manuscript does not claim algorithmic novelty (which is not inherently an issue) and the major contribution is the synthetic data pipeline and the artifact. My concerns for not recommending an acceptance lie in that I'm not fully convinced of the resulting dataset's superiority and the clarity of the data construction process.

1. **Details of the pipeline stats**: Could the authors give details about the stats of the dataset in each step (not just the final token stats for the final dataset in Table 7)? It'd be helpful to assess the bottleneck of the designed pipeline. For example, the stats of the features extracted, how many additional datapoints are generated by the step "Evolve and Merge", how many problem statements are generated (do the authors employ filtering steps? If so, the % of the filtered entries?), etc.

2. My biggest concern is how good the dataset is compared to existing code datasets: The paper only ablates on the EpiCoder dataset for the reason that they share the same synthetic nature. However, it remains unclear how the collected synthetic dataset compares to existing non-synthetic datasets such as OpenCodeReasoning. It would be good to know how large the gap if it falls behind, or if the collected dataset is even better.

Despite the authors comparing with AceReasoner 1.1 (which trains on OpenCodeReasoning) in Table 1, it uses a different backbone from what the manuscript uses, which hinders the comparison of the dataset. I will highly appreciate it if the authors could conduct SFT & RL based on the same backbone using OpenCodeReasoning, which should strengthen the claim of the paper.

3. The experiment design of the scaling effect of SFT resembles largely to AceReason-Nemotron 1.1 https://arxiv.org/abs/2506.13284, in which they also separate SFT set from v1 to v5 (growing the number of prompts) and v6 - v7 (growing the solutions per prompt) and building on top they fit a coefficient for the 2 axes for eval performance. Maybe I have missed it, but I think a reference to this is missing.

4. The first step of dual-verification sets provisional ground truth by majority vote of candidate solutions. Is there any way to get a sense of how many false positives are in the whole dataset?

5. **Reproducibility**: The authors promise to release code/data, and provide an anonymous repo link, but end-to-end reproduction of the synthetic data itself hinges on full prompt templates and sampling settings. In particular, few details are laid out for the "difficulty-weighting function".

**Questions:**

- Could the authors show how the difficulty weights are computed? L190 says "The weight w_i is determined by a set of heuristics based on input characteristics, such as character or token count, structural complexity, or semantic novelty, which serve as proxies for difficulty." What specific heuristics are used? In particular, I do not see any further mention about the "structural complexity" and “semantic novelty", how does the authors assess these quantitatively?

- I briefly checked the codebase: some prompts do not seem to be there, such as the prompt mentioned in C.2.

- For the error type analysis in Table 4, it seems that the runnable failing code is all put under Assertion Error. Is it possible to break it down into more fine-grained categories, such as Wrong Answer, Time Limit Exceeded, Memory Limit Exceeded, etc?

- Is there any plan for releasing the dataset? It's okay if it's not decided yet but would be good to know whether the community should expect an open-source dataset.

---

> ### Author Response · Authors · 2025-11-20
> **Author's Response (Part 1)**
>
> Thank you for your constructive review. You raised main concerns regarding the superiority of the proposed datasets and our open-sourcing issue.
>
> In response, we clarify dataset quality with detailed statistics, provide additional SFT+RL experiments using the Qwen2-Math backbone, add references to AceReason-Nemotron 1.1, and supplement details such as false-positive rates and fine-grained case distributions. In addition, we are fully committed to releasing our dataset, organized codebase, and trained models to the community after the rebuttal under a permissive license.
> Below, we address your concerns in detail.
>
>
> **Q1: Comparison with OpenCodeReasoning using the Same Backbone.**
>
> Under the same backbone, we achieve comparable SFT performance with significantly less data and further improve performance after RL, demonstrating the quality of our synthetic SFT and RL datasets. However, the final performance lags behind AceReason-Nemotron 1.1, and we provide an in-depth analysis of the underlying causes.
>
> During rebuttal, we have made every effort to perform SFT-then-RL training using the same backbone (Qwen2.5-Math-7B) to ensure a fair comparison with the OpenCodeReasoning dataset. The results are as follows:
>
> | Model                    | Trainset  Size                      | LCB v5 Score |
> | ------------------------ | ----------------------------- |:-------- |
> | AceReason1.1-SFT-v5      | 210k math + 218k code  | 43.3     |
> | AceReason1.1-SFT-v7      | 1.2M math + 1M code    | 51.2     |
> | AceReason1.1-RL          |     -                          | 57.2     |
> | X-Coder-Qwen-Math-7B-SFT | 200k code samples             | 50.7     |
> | X-Coder-Qwen-Math-7B-RL  | 40k code samples             | 53.6     |
>
> We achieve a comparable SFT score using significantly less data and further improve performance by an average of +2.9, demonstrating the quality of the proposed dataset from SFT to RL.
>
> However, the final model performance lags behind AceReason-Nemotron 1.1 after RL stage. We analyze three factors associated with this gap. (i) AceReason-Nemotron 1.1 achieves a solid, well-designed multi-stage RL curriculum that thoroughly explores the synergy between SFT and RL, which provides a more mature training pipeline compared with our current setup, (ii) It incorporates a rich collection of high-quality mathematical reasoning data, which has been shown to transfer effectively to code-reasoning tasks, offering a strong foundation for generalization. (iii) It leveraging high-fidelity rewards derived from verified test cases, whereas in our synthetic setting such correctness cannot be fully guaranteed.
>
> Currently, directly comparing the final performance is not particularly informative, because the data scale and RL training pipeline are not aligned at the moment. If sufficient computational resources become available, we plan to conduct a more controlled comparison with this impressive work.
>
> Besides, we believe our work remains valuable, as it provides a fully synthetic approach to improving competitive code reasoning. Real-world coding tasks are inherently limited in scale, whereas a synthetic pipeline offers far better scalability. As shown in Figure 3, our results do not exhibit a performance plateau as the synthetic dataset grows, suggesting that further gains are achievable as the data continues to scale.
>
>
> **Q2: Discussion with AceReason-Nemotron1.1.**
>
> Yes. Our SFT scaling experiment in Section 3.2, comparing “increasing the number of prompts” (v1–v4) versus “increasing solutions per prompt” (v5–v6), is directly inspired by the setup of AceReason-Nemotron 1.1. We have already cited AceReason-Nemotron 1.1 in Table 1, and we will explicitly clarify that our experimental design is inspired by them, and include proper discussion with this impressive work in the related-work section.

---

> ### Author Response · Authors · 2025-11-20
> **Authors' Response (Part 2)**
>
> **Q3: Details of the Pipeline Statistics.**
>
> We present detailed statistics on feature evolution and data filtering to demonstrate how the pipeline expands feature diversity and yields a high-quality 240k dataset.
>
> The statistics of feature extracted and evoled as follows.
> | Category                    | Features Extracted | Features After Evolution | Growth |
> | --------------------------- | ------------------ | ------------------------ | ------ |
> | **Algorithm**               | 27,400             | 176,914                  | ×6.46  |
> | **Data Structures**         | 12,353             | 65,104                   | ×5.27  |
> | **Problem Type**            | 14,134             | 130,293                  | ×9.22  |
> | **Implementation Logic**    | 12,419             | 106,157                  | ×8.55  |
> | **Complexity Analysis**     | 16,124             | 90,016                   | ×5.58  |
> | **Optimization Techniques** | 1,537              | 14,124                   | ×9.19 |
>
> The evolution strategy greatly enhances both the quantity and diversity of features, providing support for generating diverse tasks.
>
> We applied several filtering steps. Initially, we synthesize around 750k samples and perform dual verification on them.
> We then filter out tasks that cannot be solved by Gemini-2.5-Pro (i.e., tasks for which the model fails all consensus tests), which removes 35.7% of the samples.
> Among the remaining data, 8.9% of the task descriptions contain fewer than 200 tokens, and 23.4% of the tasks have golden solution that exceed the 32k context window or contain multiple Python blocks.
> After all filtering steps, we obtain 240k samples, which are then split into two non-overlapping subsets for SFT and RL.
>
> **Q4: False Positive Rate for Labeling Test Outputs via Voting.**
>
> > Is there any way to get a sense of how many false positives are in the whole dataset?
>
> On TACO-verified, we measure a 5.27% false-positive rate under voting with 8 solutions.
>
> To assess the false-positive rate of test-output labeling, we evaluate our approach on real-world, verified datasets. Specifically, we randomly sample 500 tasks from the TACO-verified dataset, and for each task we randomly retain 20 test cases.
>
> For each task, we generate n (n = 4, 8, 16) candidate solutions using R1-0528, perform majority voting on the outputs for each test input, and compare the voted consensus output against the ground-truth output to obtain a quantitative labeling accuracy. The resulting test-output labeling accuracy under different values of \(n\) is shown below.
>
>
> **Table: Average Test Output Labeling Accuracy with varying n**
> | n (# solutions) | Test Output Labeling Accuracy |
> |-----------------|-------------------------------|
> | 4               | 94.39%                        |
> | 8               | 94.73%                        |
> | 16              | 95.13%                        |
>
> **Table: Test Output Labeling Accuracy across different sources**
> | Source     | n=4    | n=8    | n=16   |
> |:---------- |:------ |:------ |:------ |
> | AtCoder    | 94.75% | 95.00% | 96.61% |
> | CodeChef   | 92.80% | 92.80% | 92.80% |
> | CodeForces | 94.44% | 94.81% | 95.06% |
>
>
> Increasing the number of sampled solutions consistently improves test output labeling accuracy. With n = 8, the false-positive rate is 5.27%, which falls within an acceptable range and demonstrates that the approach is potentially reliable to be transferred to the synthetic setting.
>
>
> **Q5: Missing Some Prompts in Attached Code.**
>
> Thank you for your interest about codebase. We prioritized the paper's core content during submission. We will refine the codebase to be organized and complete.
>
> **Q6: Difficulty Weight Details**
>
> > Could the authors show how the difficulty weights are computed?
>
> We employ two distinct strategies for assigning weights to individual test cases:
>
> 1. Semantic-Based Weighting. During test-case generation, the model is prompted to produce multiple categories of test cases (stored as .in files), including nominal (weight = 1), complex (2), boundary (3), and stress (4) scenarios. This assigns higher weights to test cases that are more likely to expose corner cases or failure modes.
> 2. Size-Based Weighting. We assign weights based on the size of the input files, which serves as a proxy for memory consumption. Specifically, we sort test cases by the size of their input files and divide them into four equal-sized buckets: the smallest 25% receive weight = 1, the next 25% receive weight = 2, the next 25% receive weight = 3, and the largest 25% receive weight = 4. This ensures that heavier test cases, which require greater memory resources, are assigned higher weights.

---

> ### Author Response · Authors · 2025-11-20
> **Authors' Response (Part 3)**
>
> **Q7: Fine-Grained Error Categorization.**
>
> > Is it possible to break down failure errors into more fine-grained categories, such as Wrong Answer, Time Limit Exceeded, Memory Limit Exceeded, etc?
>
> Following the advice, we break the runtime errors into three fine-grained categories: Wrong Answer, TLE, and MLE. The results indicate that TLE is also a major source of failure. The updated categorization is as follows:
>
> | Error Type | Qwen2.5‑Coder‑7B‑Instruct | Qwen3‑8B | X‑Coder‑7B-SFT | X‑Coder-7B |
>   | --- | --- | --- | --- | --- |
>   | Wrong Answer | 194.6 ± 10.7 | 87.1 ± 4.6 | 69.6 ± 3.7 | 67.9 ± 4.9 |
>   | Time Limit Exceeded | 18.1 ± 4.1 | 21.8 ± 3.8 | 13.7 ± 3.3 | 11.5 ± 2.6 |
>   | Memory Limit Exceeded | 0.0 ± 0.0 | 0.0 ± 0.0 | 0.0 ± 0.0 | 0.17 ± 0.38 |
>   | No Code Block Generated | 6.5 ± 8.2 | 7.7 ± 1.2 | 21.9 ± 3.7 | 11.8 ± 3.9 |
>   | Incomplete Code Block | 0.0 ± 0.0 | 0.0 ± 0.0 | 0.0 ± 0.0 | 1.0 ± 0.8 |
>   | Signature Mismatch | 0.0 ± 0.0 | 0.0 ± 0.0 | 0.0 ± 0.0 | 1.0 ± 0.8 |
>   | Syntax Error | 0.0 ± 0.0 | 0.0 ± 0.0 | 0.0 ± 0.0 | 8.3 ± 2.2 |
>
> Beyond incorrect answers, TLE (Time Limit Exceeded) is also a major issue for current Code LLMs, whereas MLE rarely occurs. This highlights the need for Code LLMs to pay more attention to code efficiency. We will update the categorization accordingly.
>
> **Q8: Codebase and Open Source Planing.**
>
> > Is there any plan for releasing the dataset?
>
> Thank you for your interest about open-sourcing. We plan to release the datasets, organized scripts, and models to the community after the rebuttal, under a MIT license.
>
> We reaffirm our commitment to open-sourcing all resources, and hope these clarifications and experiments help address your concerns. We look forward to further discussions with you.
>
> Best regards,
>
> Authors of Submission 13129

---

> ### Comment · Reviewer_vHC5 · 2025-11-20
> **Rebuttal comment**
>
> I thank the authors for the rebuttal and the efforts they put into it!
>
> Before I finalize my comment, I would like to quickly make a follow-up so that the authors could have time to respond.
>
> What I care about in my concern is "How the dataset compare with OpenCodeReasoning (or other public available code datasets)?": I would love to see a head-to-head comparison, under the same hyperparameter setting, of the main result reported in the paper SFTed with the synthetic dataset and simply replacing it with OpenCodeReasoning.
>
> I agree that comparing the final performance of the model series is not informative and I do not mean that: The authors could be using different infra and tech stack, lack of experiment details of the counterpart to compare etc.
>
> However, the ideal ablation should be: take the experiment run that ends up with the checkpoint reported in L231 (Qwen2.5-Coder-Instruct trained on your dataset), and replace the dataset with OpenCodeReasoning and re-run it again, the stopping point could be 1. finishing the same epochs. 2. elapsing the same tokens (to compensate for different token sizes, which means you need to over-epoch a bit your dataset or OpenCodeReasoning, depending on which one is smaller). It'd also be interesting to see how these ckpts behave differently when serving as a starting policy for RL on the same RL prompt set.

---

> > ### Author Response · Authors · 2025-11-21
> > **We have understood the ablation setting and are running the experiments.**
> >
> > Thank you for your recognition of our rebuttal and your timely follow-up!
> >
> > We now have a clear understanding of the desired ablation setting and are actively preparing to run the targeted experiments. We expect the results to be available in a week and will share them as soon as they are completed.
> >
> > Thank you again for your timely feedback and clarifications.

---

> > ### Author Response · Authors · 2025-11-26
> >
> > Dear Reviewer vHC5,
> >
> > We performed SFT on Qwen2.5-Coder-7B-Instruct using the OpenCodeReasoning dataset (735k samples, 6.9B tokens). To ensure a fair comparison, we trained for 4 epochs, matching the total number of training tokens in our setting (3.4B tokens × 8 epochs).
> >
> > The SFT results on LiveCodeBenchv5 are as follows:
> >
> > | Model                               | Avg. | Easy | Medium | Hard |
> > |-------------------------------------|:-------:|:----:|:------:|:----:|
> > | OCR-Qwen-7B-Instruct   | 51.3   | 95.4 | 64.0   | 18.0 |
> > | OCR-Qwen-Coder-7B-Instruct | 53.6 | 95.2 | 67.0 | 21.8 |
> > | X-Coder-Qwen-Coder-7B-Instruct      | 60.3 (+6.7) | 96.8 | 73.3 | 37.8 |
> >
> >
> > Overall, our proposed dataset yields a 6.7-point improvement after SFT, with most gains coming from the medium and hard splits. We attribute this improvement to two potential factors:
> >
> > 1. **Challenging Tasks Improve Generalization on Hard Problems.** Our pipeline synthesizes more challenging tasks, as evidenced by the much longer reasoning length in the generated solutions (average length 17.7k vs. 8.0k for OpenCodeReasoning). These challenging tasks lead to significant performance gains on the hard testset (from 21.8 to 37.8). This improvement is aligned with our observation in Figure 6 that "tasks that induce longer CoT are regarded as more valuable training data for competitive programming, as they demand deeper reasoning and are potentially more challenging."
> > 2. **Prompt Diversity Matters More Than Solution Diversity.** Our dataset contains 200k unique tasks, more than the 28.9k in OpenCodeReasoning. Expanding prompt diversity proves to be more effective than increasing solution diversity.
> >
> > Further, using OCR-Qwen-Coder-7B-Instruct as the starting point, we conduct RL on the same prompt set as in our dataset. The results are as follows:
> >
> > | Model                                      | Overall |
> > |--------------------------------------------|:-------:|
> > | OCR-Qwen-Coder-7B-Instruct-RL  | 56.0 |
> > | X-Coder-Qwen-Coder-7B-Instruct-RL          | 62.9 (+6.9) |
> >
> > After RL, OCR-Qwen-Coder-7B-Instruct further improves to 56.0, demonstrating the effectiveness of the proposed RL dataset. As our SFT model has seen a larger number of unique tasks during the SFT stage, it exhibits stronger exploration capability (pass@k), enabling it to achieve higher performance after RL.
> >
> >
> > **Table: Pass@k on LiveCodeBenchv5.**
> > | Model                         | Pass@2 | Pass@4 | Pass@8 |
> > |-------------------------------|:------:|:------:|:------:|
> > | OCR-Qwen-Coder-7B-Instruct    | 58.1   | 64.2   | 69.0   |
> > | X-Coder-Qwen-Coder-7B-Instruct| 67.9   | 73.6   | 77.0   |
> >
> > Thank you for your time and constructive comments. All of your suggestions during the rebuttal have been incorporated into the revised pdf.
> >
> > We sincerely hope that the additional ablations and clarifications can address your concerns regarding dataset superiority, and we would greatly appreciate your reconsideration of the current assessment.
> >
> > Best regards,
> >
> > The Authors

---

> > > ### Comment · Reviewer_vHC5 · 2025-11-27
> > > **Comments to rebuttal**
> > >
> > > Thank you for the efforts and the rebuttal!
> > >
> > > As I mentioned in my original review, in my point of view, the major contribution of this paper is:
> > >
> > > 1. the synthetic data pipeline
> > > 2. the artifact it produced, i.e., the dataset in the "fully synthetic" sense
> > >
> > > My main concern was that, it was not clear how the dataset compared to existing available datasets. And this concern is nicely addressed by the head-to-head comparison to OpenCodeReasoning (OCR) by the authors in the rebuttal phase and it presents a clear empirical advantage by SFTing on it. The fine-grained analysis also looks intuitive and provides hints on the potential future improvement.
> > >
> > > Despite the fact that the manuscript does not claim algorithmic novelty (which is not inherently an issue) and the training techniques are not conceptually new (SFT and RL), *I do think that the data pipeline and the dataset itself, conditioned on the empirical advantage being extensively justified, will be a solid contribution*. Therefore I encourage the authors to consolidate the experiments done here into the manuscript and better position the manuscript with existing work such as the experiment design of the scaling effect of SFT in main text, to strengthen the contribution 2.
> > >
> > > And I do hope that the authors could follow-up with efforts to further strengthen the contribution 1. such as with the commitment to release the dataset or make the whole pipeline transparent and efforts to make it easily reproducible to let the community produces their own artifact.
> > >
> > > Taking these into account I will increase my score to reflect the improvement.

---

> > > > ### Author Response · Authors · 2025-11-27
> > > >
> > > > Thank you for recognition of our contributions and for updating the score. We are also grateful for your valuable advice on continually improving our work.
> > > >
> > > > We are in total agreement regarding the two main contributions of our paper: the **synthetic data pipeline** and the resulting **synthetic artifacts**. To fairly evaluate the data’s impact, we use standard SFT and RL methods, so training algorithm novelty is not our focus. We also appreciate you suggesting this insightful experiment that highlights how the proposed dataset compared to existing available datasets, which has greatly improved the work.
> > > >
> > > > We are committed to the following:
> > > >
> > > > * **Regarding Contribution 1 (The Pipeline):** Make the entire pipeline transparent, and the codebase well-organized and easy to use.
> > > >
> > > > * **Regarding Contribution 2 (The Dataset/Experiments):** Organize experiments to ensure a better comparison with existing work to more accurately position the paper, and enable the dataset fully open-sourced. We have incorporated the additional experiments during rebuttal into the pdf and will refine them to be more complete.
> > > >
> > > > We reaffirm our commitment to releasing the artifact and the organized pipeline to the community, enabling researchers to use it directly or to generate their own artifacts.
> > > >
> > > > Finally, we are deeply grateful for your recognition and for your continued support in improving our work and supporting the research community.
> > > >
> > > > Best Regards,
> > > >
> > > > The Authors

---

### Official Review · Reviewer_pnxs · 2025-10-30

**Soundness:** 3
**Presentation:** 4
**Contribution:** 2
**Rating:** 2
**Confidence:** 5

**Summary:**

The paper presents SynthSmith, a synthetic coding-data generation pipeline designed to produce competition-level programming problems (along with test-cases and golden solutions) for training large code models. The key components of the pipeline are:
- Feature-tree generation and evolution: Starting from seed programming concepts/features, the pipeline generates and evolves feature-trees. An LLM is then tasked with selecting a suitable sub-tree (via feature‐role annotation → sub-tree selection → integration strategy) and formulating it into a natural-language task statement.
- Candidate solution + test-case generation: For each generated problem, candidate solutions and accompanying test-cases are created.
- Golden solution identification: Among the candidates, a “golden” solution is selected via a weighted scoring procedure across test-cases, forming the ground-truth entry for model training.

The resulting dataset is used to train models (the “X-Coder” family) by fine-tuning and then reinforcement learning on base models (e.g., Qwen2.5-Coder-Instruct and Qwen3-8B-Base).

**Strengths:**

- The authors describe the stages (feature-tree evolution, problem formulation, solution/test synthesis, golden-selection) in a clear and detailed manner.
- A comprehensive set of experiments was conducted on model training and performance.
- Insightful experiment design around problem style: For example, the authors examine how the style in which a problem is specified (e.g., “competitive” vs. “educational” style) affects model learning and how picking problems whose solutions require longer reasoning improved model performance relative to short-reasoning or simple difficulty-based selection.

**Weaknesses:**

**Overall**
- The idea of using concepts from seed problems and evolving them into a larger bank of problems is not entirely novel in the domain of synthetic code-data generation. The authors do not sufficiently situate their work in relation to recent pipelines such as SelfCodeAlign (Wei et al., 2024a) or CodeEvo (Sun, Qiushi et al., 2025) etc. A clear performance comparison (or final model accuracy / generation-cost comparison) with those pipelines is missing.

**Programming-Problem Generation (feature-tree → task)**
- Verification of sub-tree → task mapping: It is unclear how the authors verify that the selected sub-tree and integration strategy reliably produce reasonable programming problems (solvable by humans/LLMs, non-trivial, properly scoped). Do they evaluate the generated problem set for solvability, human-readability, or difficulty calibration? What fraction of generated problems are too ambiguous or unsolvable?
- Can an expanded feature tree give multiple integration strategies corresponding to different sub-trees? If so, are you generating multiple problems from a single feature tree?
- An ablation or experiment that simply prompts an LLM to go from feature-tree → task in one shot (without the explicit sub-tree selection/integration step) to justify the pipeline’s modular structure.

**Golden Solution Identification**
- Missing clarity on how the weights for each test case are computed (i.e., what criteria drive higher weights and by how much?)
- Your weighted-score based selection of $A'\_{golden}$ in Step-3 of Algorithm 1 does not affect Step-4. In Step-4, you are always returning the solution which has the max score on held-out validation split regardless of whether it matches the solution $A'\_{golden}$ from weighted-selection step or not (i.e. either $A\_j^+ = A\_j^* = A'\_{golden}$ or not, you are still returning $A\_j^+$). If my understanding is correct, I don’t see the point of doing weighted selection (seems superfluous).
- Further, what’s the point of assigning weights to test cases? Your selection strategy seems to implicitly allow solutions that pass some but not all test cases. Since the provisional outputs are generated using a candidate set of solutions, it ensures that there exists at least one candidate solution that passes a given test case. Beyond this point, if there exists at least one solution that passes all test-cases, we have a golden solution. However, if you are assuming that there exist programming problems for which no one candidate solution can pass all the test cases, and weights are being used to identify the solution which passes, say “more important ones”, either you are saying test cases being used are not reliable, or you can treat partially-correct solutions as golden. This may reduce the ground-truth quality. Unless the pipeline ensures fully correct solutions (pass all test-cases) this could degrade training fidelity.

**Experiments**

- The paper lacks an experiment that isolates the impact of the synthetic data generation method itself. For example: take the same base model (e.g., Qwen3-8B), fix training algorithm (SFT + RL), and compare performance when training on (a) only real data, (b) synthetic data from SynthSmith, (c) synthetic data from other pipelines. This would demonstrate whether the pipeline’s specific design yields a measurable benefit over generic synthetic methods.
- How do the authors define if a generated task is “unique”? Is uniqueness driven by selecting different sub-trees from the same feature-tree, or only by selecting entirely distinct feature-trees?
- Generality across model families: The experiments are limited to the X-Coder models derived from Qwen2.5/Qwen3.8B. It would strengthen the claim if the authors showed consistent improvements across other model families (e.g., Phi, DeepSeek, CodeLlama).
- Why do you think you are getting performance variations when the problem is being defined in different styles? Why is Leet-Code style consistently outperformed by AtCoder and CodeForces style? Can you share some insights?
- Failure-case analysis: The model trained on SynthSmith data reportedly has an increased tendency to generate no code blocks (i.e., fail to generate code) — but the authors do not provide an analysis of why this happens.
- Benchmark breadth: The evaluation focuses on one benchmark (LiveCodeBench). It lacks evaluation on other established code-benchmarks such as HumanEval+, MBPP+, EvoEval (Xia et al.), ClassEval (Du, Xueying et al.), or DS‑1000 (Lai, Yuhang et al.). This limits the external validity of the claimed improvements.

**References:**

Wei, Y., Cassano, F., Liu, J., Ding, Y., Jain, N., Mueller, Z., de Vries, H., von Werra, L., Guha, A., & Zhang, L. (2024a). SelfCodeAlign: Self-Alignment for Code Generation. Advances in Neural Information Processing Systems, 37.

Sun, Qiushi, et al. "CodeEvo: Interaction-Driven Synthesis of Code-centric Data through Hybrid and Iterative Feedback." arXiv preprint arXiv:2507.22080 (2025).

Xia, Chunqiu Steven, Yinlin Deng, and Lingming Zhang. "Top leaderboard ranking= top coding proficiency, always? evoeval: Evolving coding benchmarks via llm." arXiv preprint arXiv:2403.19114 (2024).

Du, Xueying, et al. "Classeval: A manually-crafted benchmark for evaluating llms on class-level code generation." arXiv preprint arXiv:2308.01861 (2023).

Lai, Yuhang, et al. "DS-1000: A natural and reliable benchmark for data science code generation." International Conference on Machine Learning. PMLR, 2023.

**Questions:**

See the Weaknesses Section.

---

> ### Author Response · Authors · 2025-11-20
> **Authors' Response (Part 1)**
>
> Thank you for your detailed review, as well as your positive remarks regarding the clarity of our presentation and the depth of our experimental design.
> We provide clarifications and supplementary results to address all of your concerns. We hope our rebuttal will allow you to update the ratings.
>
> **Q1: Comparison with Prior Pipelines such as SelfCodeAlign and CodeEvo.**
>
> Our method differs significantly from SelfCodeAlign and CodeEvo in terms of task focus and methodology. We target competitive-programming and construct more accurate tests to support RL training and verify solutions.
>
> (1) Task Scope. While SelfCodeAlign and CodeEvo focus on general-purpose programming, our work targets competitive programming. Competitive programming introduces distinct challenges, including synthesizing tasks that require deeper reasoning and constructing more accurate test cases. Existing approaches in this domain rely heavily on real-world data and lack scalable synthetic alternatives. In contrast, our work demonstrates that a fully synthetic pipeline can effectively support competitive programming, providing a practical and scalable solution.
>
> (2) Methodology. Our pipeline differs methodologically: we perform competition-oriented feature extraction to synthesize challenging tasks, integrate features to ensure coherent problem statements, and support multiple task styles to enhance diversity. More importantly, we construct high-fidelity test cases using the prompt-based and tool-based input generation, combined with voting-based output labeling. These accurate tests enable code RL beyond SelfCodeAlign and CodeEvo.
>
> (3) To enable a quantitative comparison, we implemented the SelfCodeAlign method and adapt it to the competitive programming setting, following the pipeline below:
>
> (i) taco seed → concepts using GPT-4o,
> (ii) concepts → instructions using GPT-o3-mini, and
> (iii) instructions → responses and tests using R1-0528, selecting successful responses that pass tests,
>
> which delivers a 10k-sample dataset with resource constraints. The average finetuning results on Qwen3-8B-Base are shown below.
>
> | Method | Task Generator | Answer Generator    | Data  | v5 Score |
> | ---------------- | -------------- | ------------------- | ----------- | -------- |
> | SelfCodeAlign    | GPT-o3-mini    | DeepSeek-R1-0528     | 10k         |   27.1       |
> | SynthSmith       | GPT-o3-mini    | DeepSeek-R1-0528     | 10k         |     31.7 (+4.6)     |
>
>
> The results indicate that the data produced by SynthSmith, benefiting from challenging problem formulation and dual verification, improves model generalization.
>
> Following your advice, we will include a discussion of SelfCodeAlign and CodeEvo in the related-work section.
>
> **Q2: Solvability of Generated Problem.**
>
> > What fraction of generated problems are too ambiguous or unsolvable.
>
> We measure the fraction of potentially unsolvable problems using GPT-5-high as the solver.
>
> To quantitatively assess the proportion of potentially unsolvable tasks, we examine the pass@1 of proprietary LLMs, including Qwen-Max, Gemini-2.5-Pro, and GPT-5-High on our voted test cases. The single-try pass rates of various proprietary LLMs are as follows:
>
> **Table: Distribution of Proprietary LLMs' Pass Rates on Voted Test Cases.**
> | Range (%)    | R1-0528 | Qwen3-Max | Gemini2.5-Pro | GPT5-High |
> |--------------|---------|-----------|----------------|-----------|
> | (0–20)       | 13.12%  | 11.06%    | 9.57%          | 3.07%     |
> | [20–40)      | 17.29%  | 16.44%    | 14.38%         | 4.83%     |
> | [40–60)      | 17.57%  | 18.59%    | 17.17%         | 6.49%     |
> | [60–80)      | 14.94%  | 16.36%    | 15.80%         | 7.80%     |
> | [80–100)     | 13.39%  | 14.39%    | 14.90%         | 10.82%    |
> | 100          | 23.66%  | 23.16%    | 28.18%         | 66.98%    |
>
> Notably, even GPT-5-High exhibits a subset of tasks with very low pass rates. Such cases can be considered potentially ambiguous, underspecified, inherently unsolvable, or affected by test labeling errors. The rationale is that if GPT-5-High fails to solve a problem, the issue is more likely due to flaws in the task itself.

---

> ### Author Response · Authors · 2025-11-20
> **Authors' Response (Part 2)**
>
> **Q3: No Guarantee of Fully Correct Golden Solutions**
>
> > Unless the pipeline ensures fully correct solutions (pass all test-cases) this could degrade training fidelity.
>
> We respectfully believe there is some misunderstanding of our work.
>
> Your concern assumes a scenario in which a fully correct, verified test set is available, and thus a golden solution that passes all test cases must exist. This assumption does not hold in the synthetic setting, where ground-truth tests are unavailable.
>
> In our pipeline, test inputs are synthesized, and test outputs are labeled by majority voting over the execution results of candidate solutions. Under this setup, a fully correct solution is not guaranteed to exist, because even the most accurate solution may fall into the minority on certain voted test cases.
>
>
> Moreover, although the selected golden solution may not pass all test cases, it remains the strongest candidate within its rollout group. Prior research has shown that imperfect solutions can still provide meaningful learning signals and effectively guide model training, as demonstrated in Figure 7 of [1] and Table 4 of [2].
>
> **Q4: Generality Across Model Families.**
>
> We supplement results on Llama-3.1-8B-Instruct to demonstrate generality beyond the Qwen series, achieving 13.4 gains after SFT and 15.3 after RL, demonstrating the quality of our dataset.
>
> Recommended DeepSeek-Coder and CodeLlama models have native 4k context windows, making them unsuitable as long CoT–reasoning foundations. We evaluate generality using Llama-3.1-8B-Instruct, which supports a 32k context window, for SFT-then-RL. The results are as:
>
> **Table: Performance on Llama-3.1-8B-Instruct**
> | Model                                 | v5 Score |
> |:------------------------------------- | -------- |
> | Llama-3.1-8B-Instruct                 | 11.8     |
> | FuseChat-Llama-3.1-8B-Instruct        | 12.6     |
> | X-Coder-Llama3.1-8B-SFT-32k-Sample    | 25.2     |
> | X-Coder-Llama3.1-8B-SFT+RL-10k-Sample |     27.1      |
>
> Given that Llama-3.1-8B-Instruct is potentially weaker than Qwen2.5-Coder-7B-Instruct in terms of code pretraining, the observed improvement from 11.8 to 25.2 to 27.1 suggests that less capable base models can also benefit from the proposed datasets.
>
>
> **Q5: Impact of Synthetic Data.**
>
> > Compare performance when training on a) only real data, b) synthetic data from SynthSmith, c) synthetic data from other pipelines.
>
> We conduct two comparisons with real-world data (AM-thinking) and synthetic data generated by SelfCodeAlign. The comparative results are shown below.
>
> **Table: Real Data vs. Data from SynthSmith**
>
> | Method | Task Generator | Answer Generator    | Data Amount | v5 Score |
> | ---------------- | -------------- | ------------------- | ----------- | -------- |
> | AM-thinking      |        -       | DeepSeek-R1-0528     | 42k         | 42.9    |
> | SynthSmith       | GPT-o3-mini    | DeepSeek-R1-0528     | 42k         | 50.0 (+7.1)    |
>
>
> **Table: Synthetic Data by SelfCodeAlign vs. Data from SynthSmith**
>
> | Method | Task Generator | Answer Generator    | Data Amount | v5 Score |
> | ---------------- | -------------- | ------------------- | ----------- | -------- |
> | SelfCodeAlign    | GPT-o3-mini    | DeepSeek-R1-0528     | 10k         |   27.1       |
> | SynthSmith       | GPT-o3-mini    | DeepSeek-R1-0528     | 10k         |     31.7 (+4.6)    |
>
> Using the same student model, Qwen3-8B-Base, our method improves over AM-thinking by an average of 7.1 points and over SelfCodeAlign by 4.6 points, indicating that the proposed pipeline provides a significant advantage compared with synthetic approaches and real-world datasets.
>
>
> **Q6: Redundancy of Weighted Selection.**
>
> > Weighted-score based selection in Step-3 of Algorithm 1 does not affect Step-4.
>
> Thank you for pointing this out. This is indeed a misalignment between the pseudo-code in Algorithm 1 and the intended procedure described in the main paper.
>
> In the current Algorithm 1, Step 3 (Weighted Selection) is redundant because the if/else logic in Step 4 incorrectly defaults to returning the hold-out validation winner in all cases. This behavior does not reflect our intended design in the paper. The correct procedure is as follows:
>
> 1. Step 3 (Weighted Selection) is our primary selection mechanism.
>    We use weighting to prioritize solutions that solve harder test cases.
> 2. Step 4 (Hold-out Confirmation) serves as a guardrail against overfitting.
>    Its role is to confirm that the weighted champion $A^{j^*}$ also performs “competitively high” on the unseen validation set.
>
> We will correct Algorithm 1 to match the intended procedure in the revised pdf.
>
> **References**
>
> [1] Liu, R.-B., et al. "Do Language Models Need to Learn Reasoning before Coding?" International Conference on Machine Learning, 2025.
>
>
> [2] Wu, J., et al. "Teaching Your Models to Understand Code via Focal Preference Alignment." Empirical Methods in Natural Language Processing, 2025.

---

> ### Author Response · Authors · 2025-11-20
> **Authors' Response (Part 3)**
>
> **Q7: Ablation on Single-Step Prompting.**
>
> > An ablation or experiment that simply prompts an LLM to go from feature-tree → task in one shot (without the explicit sub-tree selection/integration step).
>
> In short, explicit sub-tree selection is crucial as one-step approach tends to produce simpler tasks and degrade downstream performance.
>
> The rationale for the two-stage pipeline is that a single-step approach is less effective. When performing both steps simultaneously, LLMs tend to oversimplify complex instructions into trivial cases, reducing both diversity and difficulty of the generated task.
>
> To empirically validate this, we generated 32k tasks using the one-step method (feature-tree $\rightarrow$ task) and using proposed "two-stage" method (feature-tree $\rightarrow$ sub-tree $\rightarrow$ task). The SFT results on LiveCodeBench v5 are as:
>
> | Generation Method | Score (avg@4) |
> | :--- | :--- |
> | One-Step (end-to-end) | 34.8 |
> | Two-Stage (Ours) | 40.1 (+5.3) |
>
> The 5.3 gain shows that explicit sub-tree selection and integration is significantly helpful for producing high-quality, challenging tasks and justifies SynthSmith’s modular design. We will incorporate the results for clarity.
>
> **Q8: Benchmark Coverage.**
>
> Our study targets competitive programming, whereas EvoEval (program evolution), ClassEval (class implementation), and DS-1000 (data-science tasks) fall outside this scope. For completeness, we additionally report results on MBPP+ and HumanEval+, as shown below:
>
>
> | Model                         | HumanEval | HumanEval+ | MBPP | MBPP+ | Average |
> |:----------------------------- | --------- | ---------- | ---- | ----- | ------- |
> | Qwen2.5-Coder-7B-Instruct     | 88.4      | 84.1       | 83.5 | 71.7  | 81.9   |
> | X-Coder-7B-SFT                | 89.6      | 84.8       | 88.9 | 73.5  | 84.2   |
> | X-Coder-7B                    | 89.6      | 84.1       | 89.2 | 75.7  | 84.7   |
>
> We will incorporate the results into our updated version.
>
>
> **Q9: Test-Case Weighting Criteria.**
>
> > Clarity on how the weights for each test case are computed.
>
> We employ two distinct strategies for assigning weights to individual test cases:
>
> 1. Semantic-Based Weighting. During test-case generation, the model is prompted to produce multiple categories of test cases (stored as .in files), including nominal (weight = 1), complex (2), boundary (3), and stress (4) scenarios. This assigns higher weights to test cases that are more likely to expose corner cases or failure modes.
> 2. Size-Based Weighting. We assign weights based on the size of the input files, which serves as a proxy for memory consumption. Specifically, we sort test cases by the size of their input files and divide them into four equal-sized buckets: the smallest 25% receive weight = 1, the next 25% receive weight = 2, the next 25% receive weight = 3, and the largest 25% receive weight = 4. This ensures that heavier test cases, which require greater memory resources, are assigned higher weights.
>
>
> **Q10: Definition of Unique Tasks**
>
> > How do the authors define if a generated task is "unique"?
>
> Unique task is defined as sampling distinct subtrees from a shared expressive feature tree to formulate tasks.
>
> **Q11: Multiple Problems per Feature Tree.**
>
> > Are you generating multiple problems from a single feature tree?
>
> Yes. The pipeline follows a one-to-many design. We first merge all evolved features into a single, large, and expressive feature tree. From this unified tree, we repeatedly sample subtrees to formulate tasks. This design enables efficient generation of a large and diverse collection of tasks from a single, unified feature asset.

---

> ### Author Response · Authors · 2025-11-20
> **Authors' Response (Part 4)**
>
> **Q12: Performance Variations Across Styles.**
>
> The performance differences across styles are supported by the ablation results in Figure 5b rather than by heuristic assumptions. After carefully inspecting the samples of different styles, we observe that LeetCode-style problems rely on function-based starter code, whereas LiveCodeBench evaluates all styles by converting them into a unified STDIN format. This mismatch in problem format makes LeetCode-style tasks less aligned with the evaluation environment, resulting in lower performance compared with AtCoder- and CodeForces-style tasks.
>
>
> **Q13: No-Code Failure Analysis.**
>
> > The model trained on SynthSmith data reportedly has an increased tendency to generate no code blocks, and why this happens.
>
> The rise in no-code samples mainly comes from context truncation due to long cot reasoning.
>
> The base model (Qwen2.5-Coder-7B-Instruct) is a non-reasoning model that tend to generate short solution, whereas X-Coder generates long chain-of-thought reasoning. Under a 32k context window contraint, these long reasoning traces may exceed the limit and truncate before the final code, leading to more no-code outputs after SFT or RL.
>
> We carefully inspected the no-code samples and found that all of them exceeded the 32k context window, causing the reasoning process to be truncated and incomplete. We also observed that harder tasks are more likely to induce such no-code failures.
>
> X-Coder-RL generates fewer no-code outputs than its SFT counterpart, reflecting the model’s adaptation to the reward environment: no-code samples receive no reward, which discourages the model from producing such trajectories.
>
> ---
> We appreciate your time and constructive feedback and hope these clarifications and experiments help address your concerns. We look forward to further discussions with you.
>
> Best regards,
>
> Authors of Submission 13129

---

> ### Author Response · Authors · 2025-11-26
>
> Dear Reviewer pnxs,
>
> Thank you again for your helpful feedback and for taking the time to review our work.
>
> We hope our responses have addressed your concerns, and we would appreciate it if you could consider updating the score based on our clarifications. As the rebuttal deadline is approaching and the ICLR PCs have called for responses, we’re writing this message simply as a brief follow-up to see whether our clarifications resolve your questions or if anything else would benefit from further discussion.
>
> Please feel free to let us know if any further questions arise.

---

> ### Comment · Reviewer_pnxs · 2025-11-28
>
> I thank the authors of the paper for addressing my concerns, primarily regarding the novelty of the synthetic data generation pipeline, which, when used for training, shows a clear improvement in model performance, as well as the set of experiments conducted. Furthermore, the authors have clearly shown the comparison with existing real and synthetic data generation pipelines during the rebuttal phase. I will increase my score accordingly.

---

> > ### Author Response · Authors · 2025-11-28
> >
> > Dear Reviewer pnxs,
> >
> > Thank you for your recognition of our rebuttal. We are deeply grateful that your concerns regarding novelty and experimental comparison have been addressed.
> >
> > We also greatly appreciate your suggestion of the insightful experiments comparing our proposed dataset with existing real and synthetic data generation pipelines, which has improved our work in terms of novelty and empirical analysis. We have incorporated the additional results into the revised pdf.
> >
> > We see that there is currently a technical issue on the platform. If the system is restored, we would be pleased to provide a polite reminder at that time.
> >
> > Please feel free to let us know if any further questions arise, and we would be pleased to continue the discussion.
> >
> > Best Regards,
> >
> > The Authors

---

### Official Review · Reviewer_Kesq · 2025-11-04

**Soundness:** 3
**Presentation:** 3
**Contribution:** 3
**Rating:** 6
**Confidence:** 4

**Summary:**

This paper introduces SynthSmith, a fully synthetic data generation pipeline for training code reasoning models on competitive programming tasks. The pipeline generates novel tasks through feature-based synthesis, creates comprehensive test cases, produces multiple candidate solutions, and employs a dual-verification strategy to ensure quality. Based on this synthetic data, the authors train X-Coder models using both supervised fine-tuning (SFT) and reinforcement learning (RL), achieving strong performance on LiveCodeBench v5 (62.9 avg@8) and v6 (55.8 avg@8) despite having only 7B parameters. The work demonstrates that fully synthetic data can effectively train code reasoning models without relying on real-world competitive programming problems.

**Strengths:**

- The paper presents a complete pipeline covering all aspects of synthetic data generation, from tasks to solutions to test cases, with thoughtful verification strategies.
- X-Coder achieves impressive performance, outperforming larger 14B models despite having only 7B parameters, demonstrating the effectiveness of the synthetic approach.
- Extensive ablations examining verification, CoT length, task styles, data selection, and test generation methods. Detailed analysis of scaling laws showing which dimensions scale more favorably. Investigation of RL behavior including "good-gets-better" principle and resilience to noisy supervision
- Concrete insights about what makes synthetic data effective (e.g., long-CoT > short-CoT, task diversity > solution diversity)
- Section 5 provides valuable discussion of error distributions, reasoning length vs. pass rate, and undesirable patterns like reward hacking.

**Weaknesses:**

- Unclear how many tasks have incorrect "golden" solutions despite verification. The dual-verification strategy's actual error rate is not quantified
- Strong reliance on EpiCoder's feature-based framework. Significant performance gains may come from using stronger teacher models (GPT-o3-mini, Deepseek-R1-0528) rather than methodological improvements
- From Table 4, one can see that SFT or RL is increasing the number of no-code solutions from the base model, any reason why that is happening ?

**Questions:**

- Refer to weaknesses
- You mention tool-based test generation achieves 87.9% pass rate vs. 77.4% for prompting-based, but this still means 12% of test cases may be incorrect. How does test case error rate affect training?
- How do you ensure generated tasks are genuinely novel and not similar to existing competitive programming problems? Have you analyzed the diversity of generated tasks quantitatively?
- Why weren't results from all the models included for LCB v6?

---

> ### Author Response · Authors · 2025-11-20
> **Authors' Response (Part 1)**
>
> We sincerely appreciate your valuable feedback, and we would like to address each of your concerns in detail.
>
> **Q1:  Error Rate of Golden-solution.**
>
> > How many tasks have "incorrect" golden solutions despite verification.
>
> To enable quantitative assessment, we adopt two evaluations:
> (1) measuring the error rate of dual verification on our synthetic datasets, which yields pass rate distributions across various proprietary LLMs; and (2) evaluating the actual error rate on real-world datasets TACO-verified, resulting in a 7.85% error rate.
>
> (i) We first use DeepSeek-R1-0528 to generate multiple candidate solutions for each synthetic task. We then apply our dual-verification strategy to select the golden solution and measure its pass rates on the voted test cases. The pass rate is shown is below.
>
> **Table: Distribution of Golden Solution Pass Rates on Voted Test Cases using R1-0528**
> | Range (%)    | Ratio   |
> |-------------|--------|
> | (0–20)      | 13.12% |
> | [20–40)     | 17.29% |
> | [40–60)     | 17.57% |
> | [60–80)     | 14.94% |
> | [80–100)    | 13.39% |
> | 100         | 23.66% |
>
> Here, each percentage range represents the fraction of tasks whose selected golden solution attains a pass rate within that interval. For example, the [80–100) range indicates that 13.39% of tasks have golden solutions that pass between 80% and 100% of their voted test cases, while 23.66% of the solutions pass all test cases.
>
> Note that solution quality is strongly tied to model capability. The pass rates of the proprietary models (Qwen3-Max, Gemini2.5-pro, and GPT5-High) are as follows:
>
> **Table: Distribution of Proprietary LLMs' First-Try Pass Rates on Test Cases**
> | Range (%)    | Qwen3-Max | Gemini2.5-pro | GPT5-High |
> |-------------|-----------|--------|-----------|
> | (0–20)      | 11.06%    | 9.57%  | 3.07%     |
> | [20–40)     | 16.44%    | 14.38% | 4.83%     |
> | [40–60)     | 18.59%    | 17.17% | 6.49%     |
> | [60–80)     | 16.36%    | 15.80% | 7.80%     |
> | [80–100)    | 14.39%    | 14.90% | 10.82%    |
> | 100         | 23.16%    | 28.18% | 66.98%    |
>
> If we adopt a more capable model such as GPT-5-High, 66.98% of the tasks can be solved perfectly in a single attempt.
>
> (ii) We also apply our dual-verification approach to real-world, verified datasets to measure the error rate of the selected golden solutions. Because real-world datasets contain ground-truth test cases, the resulting error rate accurately reflects the true quality of the selected solutions.
>
>
> Specifically, we randomly select 500 tasks from the TACO-verified dataset, each with retained 20 test cases as ground truth tests. We apply our dual-verification procedure using R1-0528 to label test outputs via voting, and then select the golden solution based on pass rate on the voted test cases. We then evaluate each golden solution against the ground-truth tests.
>
> The verification results under different numbers of candidate solutions are as follows:
>
> | n (number of candidate solutions) | Avg. Pass Rate (test-case level) | Full Pass Rate (task-level) |
> |----------------------------------|--------------------------------------|------------------------------|
> | 4                                | 91.79%                               | 84.20%                       |
> | 8                                | 92.15%                               | 85.00%                       |
> | 16                               | 92.50%                               | 85.80%                       |
>
> On the TACO-verified dataset, our approach yields a 7.85% error rate in the selected golden solutions when n = 8. The error rate further decreases as the number of rollout solutions increases. Such an error level is acceptable, indicating that the approach has the potential to be transferred to the synthetic setting.

---

> ### Author Response · Authors · 2025-11-20
> **Authors' Response (Part 2)**
>
> **Q2: Our Contributions beyond EpiCoder.**
>
>
> Our method differs from EpiCoder in terms of task focus and methodology. We target competitive-programming tasks and produce more accurate tests to verify solutions and support RL training.
>
>
> (1) For task scope, while EpiCoder focuses on general-purpose programming, our work targets competitive programming. Competitive programming introduces challenges beyond general-purpose settings. These include generating challenging tasks that demand deeper reasoning, and synthesizing reliable tests. To tackle competitive programming, existing approaches such as AceReason1.1, and Olympic-Coder rely heavily on real-world data and lack a more scalable approach. In contrast, our work demonstrates that a fully synthetic pipeline can effectively address competitive programming, offering a scalable and practical solution.
>
> (2)  For methodology, by comparison, we introduced three key improvements beyond EpiCoder: competition-oriented feature extraction to formulate challenging tasks, dedicated feature integration to enhance task coherence, and multi-style task construction to enrich diversity (L140). Competitive tasks demand highly accurate test cases for solution verification and RL training. To tacke this, we introduce the prompting- and tool-based test input generation along with the voting strategy to label test output, yielding more reliable test cases. These test cases enable code RL, which is not supported by EpiCoder.
>
> Empirically, using the same teacher model R1-0528, we achieve a 21% absolute improvement over EpiCoder (Figure 5c), isolating the gain from the stronger teacher.
>
>
> **Q3: LCB v6 Results.**
>
> > Why weren't results from all the models included for LCB v6?
>
> The v6 results for all models were not included because some models, such as rStar-Coder and AceReason1.1-7B-SFT, are unavailable. We supplement the LCB v6 results as follows and will update the Table 1 accordingly.
>
> | Model              | Size | LCB v6 Score |
> | ------------------ | ---- | ------------ |
> | X-Coder-7B-RL            | 7B   | 55.8         |
> | X-Coder-7B-SFT        | 7B   | 53.5         |
> | X-Coder-8B-RL            | 8B   | 56.5         |
> | X-Coder-8B-SFT        | 8B   | 55.4         |
> | Mimo-7B-RL            | 7B   | 49.3         |
> | Mimo-7B-SFT           | 7B   | 45.5         |
> | AceReason1.1-7B    | 7B   | 52.1         |
> | Klear-Reasoner-8B  | 8B   | 53.1         |
> | Klear-Reasoner-8B-SFT | 8B   | 49.6         |
> | Qwen3-8B              | 8B   | 48.4         |
> | OpenThinker3-7B       | 7B   | 40.8         |
> | OCR-Qwen-Instruct-7B  | 7B   | 44.5         |
> | Skywork-OR1-7B      | 7B   | 40.0         |
> | OlympicCoder-7B       | 7B   | 19.3         |
> | Bespoke-Stratos-7B    | 7B   | 8.57         |
> | DeepCoder-14B-Preview       | 14B  | 48.5         |
> | AReal-boba²-14B     | 14B  | 56.7         |
>
>
> **Q4: How does Test Case Error Rate Affect Training.**
>
> Occasional test-case errors do not significantly degrade RL performance due to continuous rewards and GRPO’s robustness.
>
> For reward design, we adopt a continuous reward that reflects the fraction of passed tests rather than an all-or-nothing signal. As a result, a few mislabeled cases introduce only minor noise, and the model still receives a valid learning signal from the majority of correctly labeled tests. As shown in Section 3.3, the RL stage remains stable under such imperfections.
>
> For optimization algorithm, GRPO relies on relative advantage rankings with group. It does not require every test case to be perfectly labeled. When mislabeled cases affect all candidate solutions similarly, the resulting noise is mitigated in the relative comparison, making the algorithm resilient to imperfect or noisy tests.
>
> Empirically, synthetic RL yields meaningful gains, improving X-Coder-SFT-7B from 60.3 to 62.9 averagely, showing that it can learn under a potentially noisy reward signal.

---

> ### Author Response · Authors · 2025-11-20
> **Authors' Response (Part 3)**
>
> **Q5: Diversity of Generated Tasks.**
>
> We quantify task diversity through embedding-based cluster analysis and find that our dataset exhibits higher diversity than the Evol-Instruct-Code dataset.
>
> We embed all tasks using jina-embeddings-v2-base-code, project them into 2D using t-SNE, cluster the projected points with K-means (K=10), and compute the average Euclidean distance between cluster centroids as the diversity metric.
>
> In our datasets (randomly sampled 10k), cluster sizes range 529-1,612 items, average centroid distance 0.613. In the Evol-Instruct-Code dataset, the average is 0.507. Larger average euclidean distances indicate our dataset show greater diversity than Evol-Instruct-Code.
>
> We will include the t-SNE visualization results in the revised pdf.
>
> **Q6: Explanation for No-Code Samples.**
>
> >SFT or RL is increasing the number of no-code solutions from the base model, any reason why that is happening?
>
> The rise in no-code samples mainly comes from context truncation due to long cot reasoning.
>
> The base models are non-reasoning model that tend to produce short solutions, whereas trained X-Coder generates long chain-of-thought reasoning. Under a 32k context window, these long reasoning traces may exceed the limit and truncate before the final code, leading to more no-code outputs.
>
> We carefully inspected the no-code samples and found that all of them exceeded the 32k context window, causing the reasoning process to be truncated and incomplete.
>
> We appreciate your feedback and hope we can address your concerns. Feel free to contact us if you have further questions.
>
> Best regards,
>
> Authors of Submission 13129

---

> ### Author Response · Authors · 2025-11-26
>
> Dear Reviewer Kesq,
>
> Thank you again for your helpful feedback and for the time to review our work.
>
> We hope our responses have addressed your concerns, and we have incorporated corresponding suggestions into our revised pdf. We are writing this message simply as a brief follow-up to see whether our clarifications resolve your questions or if anything else would benefit from further discussion. Please feel free to let us know if any further questions arise, and we would be glad to provide additional information or run experiments before the discussion period ends.
>
> Best Regards,
>
> The Authors

---

### Author Response · Authors · 2025-11-30
**Summary of the Work and Rebuttal for the Area Chair**

Dear Area Chair,

Thank you so much for taking the additional time to review our paper. We are writing to provide a brief summary of our work and the rebuttal for your convenience.

**Work Summary:** To address persistent data scarcity in competitive programming, we aim to build a scalable alternative by training Code LLMs with fully synthetic data. Building on this motivation, our paper makes two connected contributions: **(i) a synthetic data pipeline** for competitive programming, capable of generating diverse, challenging tasks along with verified solutions and test cases; and **(ii) a 240k-sample synthetic dataset** to support both SFT and Code RL. Together, the pipeline and dataset demonstrate that high-quality synthetic data can significantly advance code reasoning capabilities.

Before the rebuttal, the work was recognized for its "impressive performance" (Reviewer Kesq, vHC5), comprehensive experiments (Reviewer Kesq, pnxs, vHC5), and "concrete insights about what makes synthetic data effective" (Reviewer Kesq, pnxs).

**Rebuttal Updates:** During the rebuttal, we mainly addressed the following concerns: **(i) Comparison with existing pipelines** (from Reviewer Kesq [W2], Reviewer pnxs [Overall]): We thoroughly compared our pipeline with existing approaches to distinguish it in terms of task scope and methodological details. **(ii) Comparison with existing datasets** (from Reviewer vHC5 [W2], Reviewer pnxs [Experiments]): We supplemented a head-to-head comparison with OpenCodeReasoning dataset to demonstrate the clear empirical advantage of our dataset. **(iii) Clarifications:** We clarified the failure analysis and supplemented the pipeline statistics, and incorporated these details into the revised pdf.

**Current Status:** Authors and reviewers engaged actively throughout the rebuttal. We are pleased to see consistently positive feedback. **(i) Reviewer vHC5** raised the score to **6**, acknowledging that "the concern is nicely addressed by the head-to-head comparison to OpenCodeReasoning and intuitive analysis." We are particularly encouraged by the reviewer's conclusion that "I do think that the data pipeline and the dataset itself, conditioned on the empirical advantage being extensively justified, will be a solid contribution."; **(ii) Reviewer pnxs** explicitly stated that they '**will increase the score accordingly**' for all concerns resolved. **(iii) Reviewer Kesq** has maintained the original positive score of **6** to date. Since no further concerns were raised, we did not post additional responses. We sincerely thank the reviewers for the constructive feedback, which greatly helped us improve the paper.

Given that the pipeline and dataset can significantly advance code reasoning capabilities, we reaffirm our commitment to releasing the artifacts and the organized pipeline to the community, enabling researchers to use them directly or generate their own artifacts.

Best regards,

The Authors of Submission 13129

---

### Meta-Review · Area_Chair_3QRD · 2026-01-23

**Summary:**

The paper develops a synthetic training pipeline for code reasoning models and applies the pipeline to competitive programming. There is no fundamental algorithmic novelty; the primary contribution is the artifact.

The reviewers found the overall approach reasonable; the main concerns were about novelty and missing comparisons with methods such as SelfCodeAlign, CodeEvo, and OpenCodeReasoning. During the response period, the authors supplied additional data showing wins over these baselines. This was moderately persuasive, but no reviewer was willing to champion the paper in the end. My personal opinion is that considering the large amount of related work in this space, it would be desirable for the authors to create a new version with the additional results and send it through another round of peer review. Given this, I am recommending rejection.

**Reviewer Concerns:**

See the reviews.

**Reviewer Scores:**

Two of the reviewers agreed to increase their scores. However, the comments do not indicate a level of enthusiasm so strong that the paper would have been a clear accept.

---

### Decision · Program_Chairs · 2026-01-26

Reject